# Intracellular niche-specific profiling reveals transcriptional adaptations required for the cytosolic lifestyle of *Salmonella enterica*

**TuShun R. Powers**[1☯], **Amanda L. Haeberle**[1☯], **Alexander V. Predeus**[2], **Disa L. Hammarlöf**[2¤], **Jennifer A. Cundiff**[1], **Zeus Saldaña-Ahuactzi**[1], **Karsten Hokamp**[3], **Jay C. D. Hinton**[2], **Leigh A. Knodler**[1]*

**1** Paul G. Allen School for Global Health, College of Veterinary Medicine, Washington State University, Pullman, Washington, United States of America, **2** Institute of Infection, Veterinary and Ecological Sciences, University of Liverpool, Liverpool, United Kingdom, **3** Smurfit Institute of Genetics, School of Genetics and Microbiology, Trinity College Dublin, Dublin, Ireland

☯ These authors contributed equally to this work.
¤ Current address: SciLifeLab, Science for Life Laboratory, KTH, Sweden
* leigh.knodler@wsu.edu

**Data Availability Statement:** Raw sequencing reads have been deposited at Gene Expression Omnibus (GEO) database under series ID

## Abstract

*Salmonella enterica* serovar Typhimurium (*S.* Typhimurium) is a zoonotic pathogen that causes diarrheal disease in humans and animals. During salmonellosis, *S.* Typhimurium colonizes epithelial cells lining the gastrointestinal tract. *S.* Typhimurium has an unusual lifestyle in epithelial cells that begins within an endocytic-derived *Salmonella*-containing vacuole (SCV), followed by escape into the cytosol, epithelial cell lysis and bacterial release. The cytosol is a more permissive environment than the SCV and supports rapid bacterial growth. The physicochemical conditions encountered by *S.* Typhimurium within the epithelial cytosol, and the bacterial genes required for cytosolic colonization, remain largely unknown. Here we have exploited the parallel colonization strategies of *S.* Typhimurium in epithelial cells to decipher the two niche-specific bacterial virulence programs. By combining a population-based RNA-seq approach with single-cell microscopic analysis, we identified bacterial genes with cytosol-induced or vacuole-induced expression signatures. Using these genes as environmental biosensors, we defined that *Salmonella* is exposed to oxidative stress and iron and manganese deprivation in the cytosol and zinc and magnesium deprivation in the SCV. Furthermore, iron availability was critical for optimal *S.* Typhimurium replication in the cytosol, as well as *entC*, *fepB*, *soxS*, *mntH* and *sitA*. Virulence genes that are typically associated with extracellular bacteria, namely *Salmonella* pathogenicity island 1 (SPI1) and SPI4, showed increased expression in the cytosol compared to vacuole. Our study reveals that the cytosolic and vacuolar *S.* Typhimurium virulence gene programs are unique to, and tailored for, residence within distinct intracellular compartments. This archetypical vacuole-adapted pathogen therefore requires extensive transcriptional reprogramming to successfully colonize the mammalian cytosol.

GSE179103. All other relevant data is within the manuscript and its Supporting Information files.

**Funding:** This work was supported by a Wellcome Trust Senior Investigator award (Grant 106914/Z/15/Z; https://wellcome.org) to JCDH; NIH NIAID (https://www.niaid.nih.gov/) T32 training grant AI007025 to TRP; start-up funds provided by the Paul G. Allen School for Global Health and grants from WSU College of Veterinary Medicine (Mr. and Mrs. Delbert Caldwell Endowment and USDA NIFA Animal Health and Disease Funds) and Burroughs Wellcome Fund (https://www.bwfund.org.org/) to LAK. LAK holds an Investigators in the Pathogenesis of Infectious Disease Award from the Burroughs Wellcome Fund. The funders had no role in study design, data collection and analysis, decision to publish, or preparation of the manuscript.

**Competing interests:** The authors have declared that no competing interests exist.

## Author summary

Intracellular pathogens reside either within a membrane-bound vacuole or are free-living in the cytosol and their virulence programs are tailored towards survival within a particular intracellular compartment. Some bacterial pathogens (such as *Salmonella enterica*) can successfully colonize both intracellular niches, but how they do so is unclear. Here we have exploited the parallel intracellular lifestyles of *S. enterica* in epithelial cells to identify the niche-specific bacterial expression profiles and environmental cues encountered by *S. enterica*. We have also discovered bacterial genes that are required for colonization of the cytosol, but not the vacuole. Our results advance our understanding of pathogen-adaptation to alternative replication niches and highlight an emerging concept in the field of bacteria-host cell interactions.

## Introduction

There are two major niches in which intracellular bacteria survive and proliferate after internalization into host cells, confined within a membrane-bound vacuole or free-living within the cytosol. Different pathogenic mechanisms are required to occupy these diverse environments. While bacterial pathogens have been historically categorized as being either vacuolar or cytosolic, it has recently been realized that some bacteria can occupy both niches, often in a cell-type specific manner. Examples are *Salmonella enterica*, *Mycobacterium tuberculosis* and *Listeria monocytogenes* [1–3]. What is unclear is how bacteria that are adapted to survive within one intracellular niche can successfully colonize a distinct cellular compartment.

Of the foodborne bacterial, protozoal and viral diseases, non-typhoidal *Salmonella enterica* (NTS) cause the largest burden of illness and death worldwide [4]. Infection can cause either a self-limiting gastroenteritis or a life-threatening, invasive disease (invasive non-typhoidal salmonellosis) in immunocompromised individuals, which is particularly a public health problem in sub-Saharan Africa and south-east Asia [5]. Upon ingestion of contaminated food or water, *S. enterica* enters epithelial cells lining the gut and resides within a membrane-bound compartment derived from the endocytic pathway [6,7], the *Salmonella*-containing vacuole (SCV). Entry into non-phagocytic cells is largely governed by a type III secretion system, T3SS1, that is encoded by *Salmonella* pathogenicity island (SPI) 1. The T3SS acts as a molecular syringe to inject bacterial effector proteins across the eukaryotic cell plasma membrane to modulate host signaling networks that induce rearrangement of the actin cytoskeleton, leading to the formation of plasma membrane "ruffles" and engulfment of bacteria (reviewed in [8]). Establishment and maintenance of the SCV is dependent upon effectors being translocated across the vacuolar membrane by T3SS1 and another T3SS, T3SS2, which is specifically induced by environmental cues sensed by bacteria within the SCV lumen [9,10]. T3SS2 is also important for survival within phagocytic cells, which *S. enterica* encounters in the lamina propria during an enteric infection or in the mesenteric lymph nodes, reticuloendothelial tissues (liver and spleen) and circulating blood during invasive disease.

*S. enterica* resides within a membrane-bound vacuole within epithelial cells, fibroblasts, macrophages and dendritic cells (reviewed in [11]). However, the intracellular lifestyle of *S. enterica* differs between cell types, with a proportion of the total bacterial population living freely in the cytosol of epithelial cells, a phenomenon described for *Salmonella enterica* serovar Typhimurium (*S.* Typhimurium) and *S.* Typhi infections *in vitro* and/or *in vivo* [12–17]. The eventual outcome of epithelial cells harboring cytosolic bacteria is their expulsion from the monolayer into the lumen of the gut and gall bladder [12,16–19], serving as a mechanism for

bacterial spreading within and between hosts. Notably, the cytosol of macrophages is not permissive for *S.* Typhimurium replication [20,21], possibly due to nutrient limitation or enhanced host cell innate immune defenses such as autophagy and/or inflammasomes [21]. Considering that the site of intracellular replication is cell-type restricted, we propose that *S. enterica* is an "opportunistic" cytosolic pathogen.

A specialized form of autophagy, called xenophagy, protects the mammalian cytosol by targeting intracellular pathogens to autophagosomes for their eventual degradation in lysosomes (reviewed in [22]). In fact, many of the components of the autophagic machinery have been identified using *S.* Typhimurium as a model pathogen. In the first report of autophagic recognition, Brumell and colleagues showed that a sub-population of internalized *S.* Typhimurium damage their nascent vacuole in epithelial cells in a T3SS1-dependent manner and these bacteria are decorated with the autophagy marker, microtubule-associated light chain-3 (LC3) [23]. A recently identified type III effector, SopF, limits the decoration of SCVs with LC3 [24,25]. Damage exposes host glycans restricted to the vacuole lumen to the cytosol, which are then recognized by β-galactoside-binding lectins, specifically galectin-3 (GAL3), GAL8 and GAL9 [26]. Galectin binding acts as an "eat me" signal that initiates a cascade of receptor binding, phagophore formation and tethering to the bacterium, culminating in autophagosome formation [26,27]. However, autophagic control of *Salmonella* is temporal and incomplete, at least in epithelial cells [12,23]. Furthermore, *S.* Typhimurium can also use autophagy to promote replication in the cytosol of epithelial cells [28] and repair damaged SCVs in mouse embryonic fibroblasts [29]. Autophagy therefore serves both a pro- and anti-bacterial role in *S. enterica* infections.

Previous studies have largely investigated the infectious cycle of *S.* Typhimurium in epithelial cells by population-based analyses, which do not account for the heterogeneous population of intracellular bacteria. Only by determining the distinct responses of intracellular *Salmonella* to the cytosolic and the vacuolar niche can the infection biology of this important pathogen be accurately defined. Whilst the distinct milieus encountered within a vacuole versus the cytosol provide site-specific cues for *Salmonella* gene induction, little is known about what these cues might be, or which bacterial genes drive replication/survival in the cytosol. Using a combination of RNA-seq-based transcriptomics and single-cell microscopic analysis, here we describe niche-specific environments encountered by *S.* Typhimurium in epithelial cells and identify *S.* Typhimurium genes that are required for bacterial proliferation in the cytosolic compartment.

## Results

### Modulation of autophagy affects bacterial proliferation in the epithelial cell cytosol

We hypothesized that the modulation of autophagy in epithelial cells would selectively perturb the cytosolic, and not vacuolar, proliferation of *S.* Typhimurium. Autophagy levels can be manipulated by pharmacological or genetic means (reviewed in [30]). To test our hypothesis, we used nutrient starvation to upregulate autophagy and the class III phosphoinositide 3-kinase (PI3K) inhibitor, wortmannin (WTM), to inhibit autophagy (Fig 1A). To enumerate cytosolic bacteria after autophagy activation/inhibition, we used the digitonin permeabilization assay [12,13] to label the bacteria accessible to cytosol-delivered anti-*S.* Typhimurium lipopolysaccharide (LPS) antibodies at the early stages of infection (15 min– 3 h p.i., Fig 1B). Wild-type bacteria were constitutively expressing *mCherry* from a plasmid, pFPV-mCherry. In untreated cells (basal levels of autophagy), 6.8% of bacteria were accessible to the cytosol as early as 15 min p.i. This proportion increased to ~20% by 45 min p.i. and remained at a steady-state thereafter (Fig 1B). Treatment with Earle's balanced salt solution (EBSS), i.e.

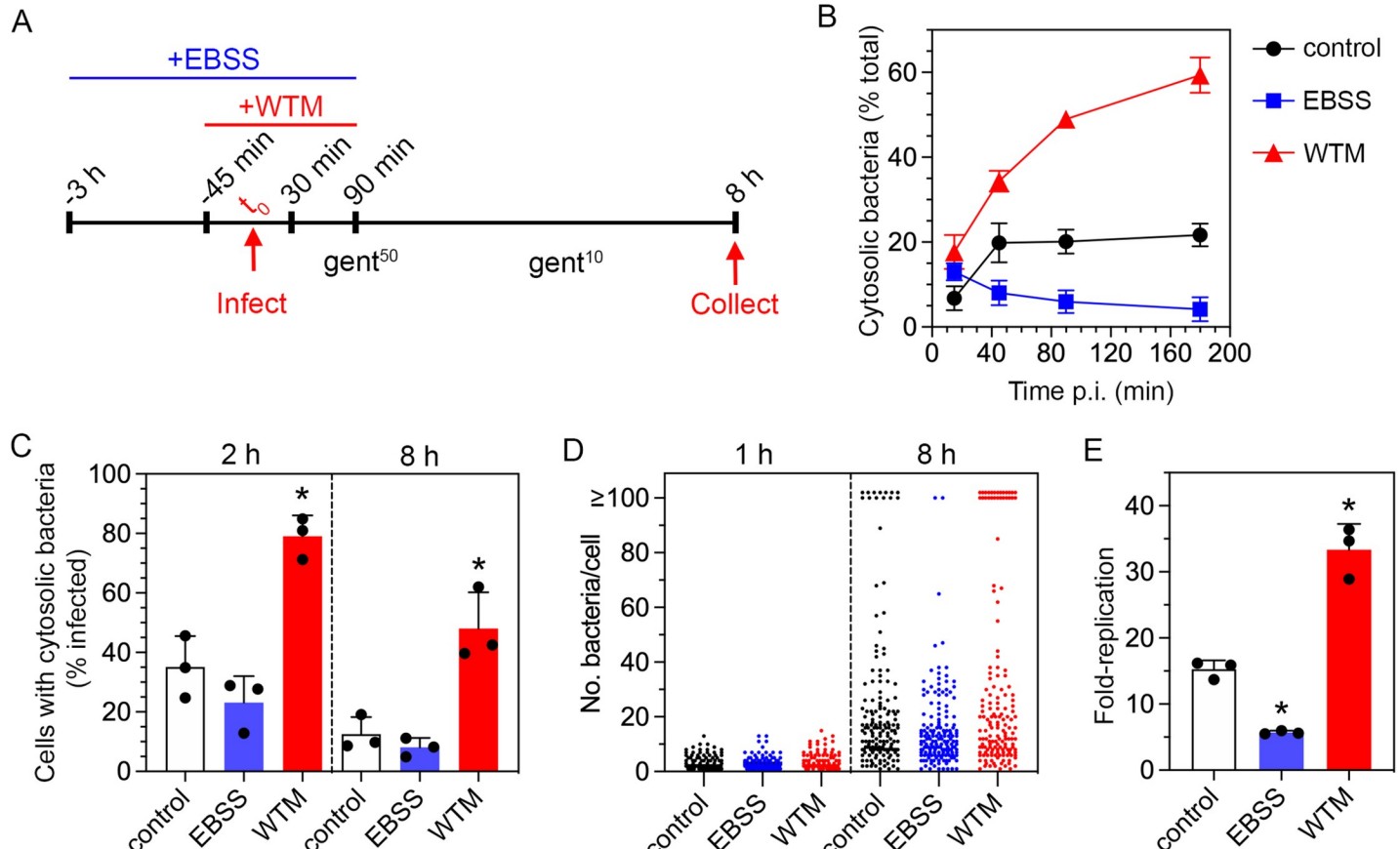

**Fig 1. EBSS and wortmannin inversely affect the cytosolic population.** (A) Schematic of the experimental design employed to differentially modulate host cell autophagy. (B) Epithelial cells seeded on glass coverslips were pretreated with EBSS or 100 nM WTM as depicted in (A) and infected with *S*. Typhimurium wild-type bacteria harboring pFPV-mCherry for plasmid-borne, constitutive expression of *mCherry*. The proportion of bacteria accessible to anti-*Salmonella* LPS antibodies delivered to the mammalian cell cytosol was determined by digitonin permeabilization assay and fluorescence microscopy. n≥3 independent experiments. (C) Epithelial cells seeded on coverslips were pretreated as in (A) and infected with mCherry-*S*. Typhimurium (wild-type *glmS*::P*trc-mCherryST*::FRT bacteria constitutively expressing a chromosomal copy of *mCherry*) harboring the fluorescent reporter plasmid, P*uhpT-gfpova* (pNF101). At the indicated times, cells were fixed and the proportion of infected cells containing cytosolic bacteria (GFP-positive) was scored by fluorescence microscopy. The mean from each experiment is represented as a large dot (n = 3). (D) Cells seeded on coverslips were pretreated as in (A) and infected with wild-type bacteria harboring pFPV-mCherry. The number of bacteria in each infected cell was scored by fluorescence microscopy. Cells with ≥100 bacteria contain cytosolic *S*. Typhimurium. Each small dot represents one infected cell. n = 2 (1 h) or 3 (8 h) experiments. (E) Epithelial cells were infected with wild-type bacteria and the number of intracellular bacteria at 1 h and 8 h p.i. was determined by gentamicin protection assay. Fold-replication is CFUs at 8 h/1 h. The mean from each experiment is represented as a large dot (n = 3). For all panels, control = untreated cells; EBSS = Earle's balanced salt solution treatment; WTM = wortmannin treatment. Asterisks indicate data significantly different from control (ANOVA with Dunnett's post-hoc test, p<0.05).

starvation-induced autophagy, reduced the fraction of bacteria that were accessible to the cytosol at 45 min, 90 min and 180 min p.i. (Fig 1B), consistent with enhanced autophagic capture of bacteria in, or repair of, damaged vacuoles. In contrast, inhibition of autophagy (WTM) increased the proportion of cytosolic bacteria at all timepoints examined (Fig 1B), in agreement with previous findings [14,31].

To verify cytosolic localization, we used a transcriptional reporter plasmid, pNF101, that only expresses *gfp-ova* in the sub-population of *S*. Typhimurium that are in damaged vacuoles and/or free in the cytosol. Epithelial cells were infected with mCherry-*S*. Typhimurium (*S*. Typhimurium wild-type bacteria constitutively expressing *mCherry* on the chromosome) harboring pNF101. In untreated cells, 35% and 13% of infected cells harbored GFP-positive (cytosolic) bacteria at 2 h and 8 h p.i., respectively (Fig 1C). EBSS treatment lowered this proportion

at both timepoints to 23% and 8%, respectively. Conversely, WTM treatment increased the level of infected cells with cytosolic bacteria at both 2 h and 8 h p.i. (79% and 48%).

To quantify the niche-specific effects of EBSS and WTM treatments on bacterial replication, epithelial cells were infected with *S*. Typhimurium harboring pFPV-mCherry and the number of bacteria per cell was scored. Cytosolic *S*. Typhimurium that evade autophagic clearance in epithelial cells replicate at a much faster rate than vacuolar bacteria [12,13,32], eventually occupying the entire cytosolic space ($\geq$100 bacteria/cell). By contrast, vacuolar bacteria display low to moderate replication (2–40 bacteria/cell). Comparing the three treatment conditions, the incidence of infected cells with $\geq$100 bacteria at 8 h p.i. (8.2±1.6% in control, n = 3 experiments) was reduced by EBSS treatment (1.2±2.0%) and promoted by WTM treatment (16 ±5.4%) (Fig 1D). A similar trend was seen with the chloroquine (CHQ) resistance assay, which quantifies the proportion of cytosolic bacteria in the total population—51±8.7%, 28±14% and 65±10% (n = 4 experiments) of intracellular bacteria were cytosolic at 8 h p.i. in untreated, EBSS-treated and WTM-treated cells, respectively. Consequently, total bacterial proliferation was restricted and promoted by EBSS and WTM treatment, respectively, compared to untreated cells (Fig 1E). Importantly, neither bacterial invasion at 1 h p.i. nor vacuolar replication at 8 h p.i. was overtly affected (Fig 1D). Collectively, these data confirm that modulation of autophagy selectively affects the proliferation of *S*. Typhimurium in the cytosol of epithelial cells.

## Identification of cytosol- and vacuole-induced genes by RNA-seq analysis

To define the transcriptional signature of cytosolic *S*. Typhimurium, we isolated bacterial RNA from epithelial cells that had been treated with EBSS or WTM, conditions that depleted or enriched the cytosolic population, respectively. Treated cells were infected with wild-type bacteria (Fig 1A), samples were collected and processed at 8 h p.i. and bacterial RNA was extracted using an enrichment protocol described previously ([33]; S1 Fig). Importantly, EBSS and WTM treatments were restricted to the early stages of infection ($\leq$90 min p.i.) (Fig 1A) to limit any effects on bacterial gene expression at the time of sample collection (8 h p.i.). Four cDNA libraries were derived from two biological replicates and analyzed by RNA-seq (see Materials and Methods; [34]). The entire dataset is available in S1 Dataset and includes the expression profiles of the >5,000 coding and non-coding genes identified previously in *S*. Typhimurium [34,35]. Genes for which the adjusted p-value was 0.05 or less (DeSeq2 analysis) were deemed as preferentially induced in the cytosol or vacuole, resulting in a shortlist of 216 "up cytosol" genes and 443 "up vacuole" *S*. Typhimurium genes (Fig 2A and S1 Dataset).

Of the twelve chromosomally-encoded pathogenicity islands (PAIs) in *S*. Typhimurium, only four were up-regulated in the cytosolic environment (S2 Fig). Most obvious was the abundance of SPI1 genes that were up-regulated, consistent with the reported induction of *prgH* in the cytosolic population at later times in epithelial cells [12,36]. SPI4, which encodes a giant non-fimbrial adhesin and its cognate type I secretion system and is co-regulated with SPI1 [37,38], was also up-regulated in the cytosol. Lastly, genes encoding for two T3SS1 effectors within SPI5 and SPI11, SopB [39,40] and SopF [24,25,41], respectively, were up-regulated in cytosolic bacteria. The remaining SPI5 and SPI11 genes were either up-regulated in the vacuolar population, or had unchanged expression, highlighting the mosaic nature of PAIs. Most SPI2 genes were up-regulated in the vacuolar population, as were genes encoding two type III effectors translocated by T3SS2 (*pipB* in SPI5 and *sspH2* in SPI12). SPI3 genes were up-regulated in the SCV of epithelial cells, like in macrophages [42,43]. Similarly, most genes within SPI11, a PAI that is important for macrophage survival [44], and SPI12 were up-regulated in the epithelial SCV. The majority of SPI6 genes (which encode a type VI secretion system;

A

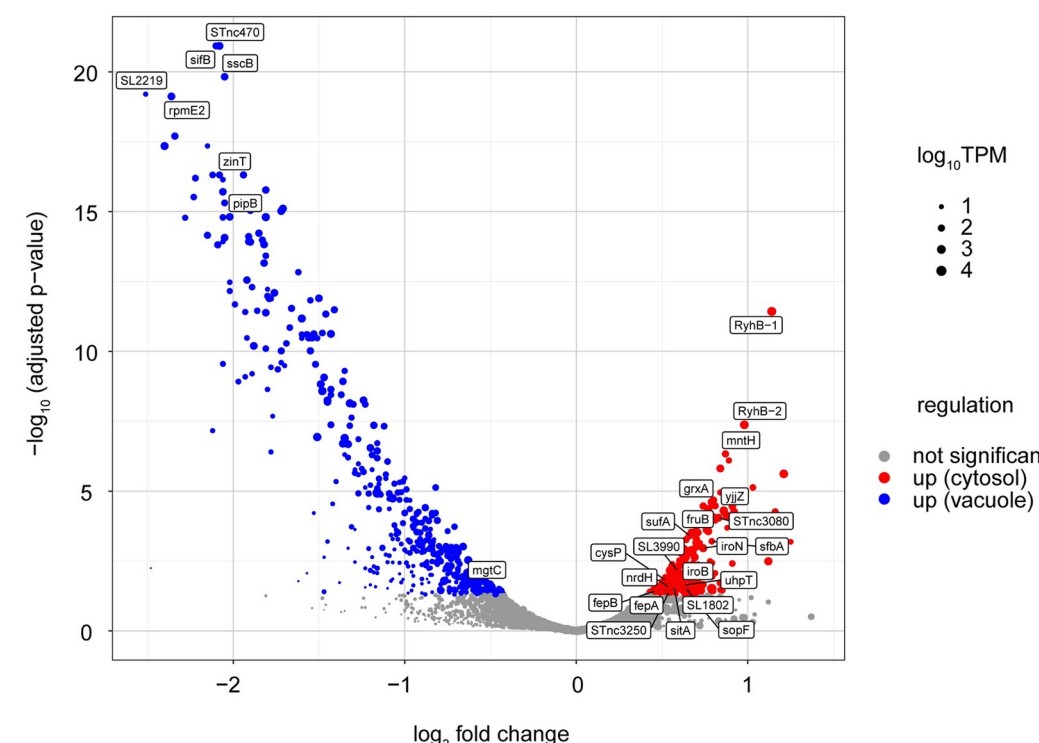

B

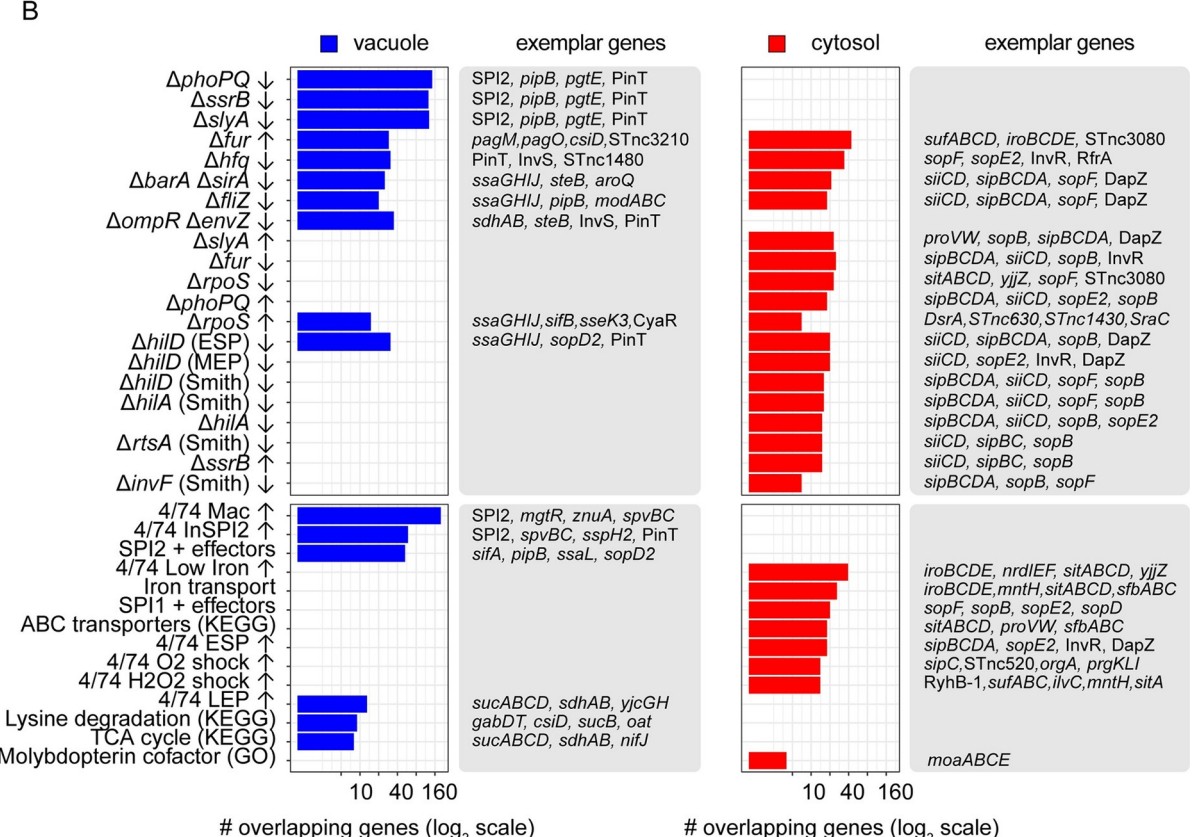

**Fig 2. Summary of RNA-seq results.** (A) RNA-seq results displayed as a volcano plot. Statistical significance is represented on the y-axis and magnitude of change on the x-axis. Statistically significant mRNAs and sRNAs are shown in blue dots for "up vacuole" and red dots for "up cytosol" categories. mRNAs/sRNAs that did not pass statistical significance are indicated by grey dots. Select genes of interest are indicated. (B) Pathway enrichment analysis. Top panels indicate regulons generated from transcriptional profiling of key regulatory mutants, while bottom panels display enriched gene sets generated from KEGG, GO, or from RNA-seq profiling of *S*. Typhimurium under 22 infection-relevant *in vitro* growth conditions. All displayed enrichments are significant (adjusted p-value <0.05) according to Fisher's exact test with Benjamini-Hochberg correction for multiple comparisons. The number of genes in each category is shown in $\log_2$ scale. Only a selection of non-redundant pathways is presented; the complete list is available as S2 Dataset.

[45]), all SPI9 genes (encode a type I secretion system; [46]) and SPI16 genes were not expressed in bacteria residing within either niche in epithelial cells. Overall, the transcriptional signatures of cytosolic and vacuolar bacteria at the PAI level (S2 Fig) validates the basis of our RNA-seq analysis.

To put the observed gene expression changes into the context of the extensive *S. enterica* literature, we used an innovative approach that involved the curation of 80 custom pathways from previously published regulons that had been defined by RNA-seq, ChIP-seq or microarray-based analysis. We combined these pathways with KEGG and GO gene sets to annotate the subsets of "up cytosol" and "up vacuole" *S*. Typhimurium genes (Fig 2B and S2 Dataset). The list of "up vacuole" genes included an abundance (e.g. SPI2, *pipB*, *pgtE*, PinT, InvS, *steB*) that are positively regulated by the two-component systems that govern SPI2 induction in the SCV, namely PhoPQ, SsrAB and OmpRZ. Conversely, genes that are negatively regulated by these two component systems were in the "up cytosol" category. The SPI1-encoded transcription factors, HilA, HilD, and InvF, are required for the expression of SPI1 genes [47], and accordingly many of the "up cytosol" genes/sRNAs (e.g. *sipBCDA*, *siiCD*, *sopF*, *sopB*) grouped with these regulons. Our analysis also revealed several specific gene signatures (Fig 2B). For example, genes associated with iron transport were solely up-regulated in the cytosol. Up-regulation of the tricarboxylic acid (TCA) cycle, lysine degradation and molybdate transport was evident in vacuolar bacteria, suggesting increased reliance of *Salmonella* on these energy sources in the SCV. Interestingly, genes/sRNAs that are repressed by Ferric uptake regulator (Fur) in iron-replete conditions and activated by Hfq, BarA/SirA and FliZ were up-regulated in both the vacuole and cytosol.

We used the expression profiles of *S*. Typhimurium grown in 22 distinct infection-relevant *in vitro* growth conditions [34] to predict the environmental conditions experienced by *S*. Typhimurium when in the cytosol and vacuole. For the 216 genes up-regulated in the cytosol, three main patterns emerged: (i) up-regulation by bile shock (3% bile for 10 min) and/or low $Fe^{2+}$ shock (200 μM 2,2-dipyridyl (DPI) for 10 min) (e.g. *mntH*, *sitABCD*, *entCEBA*), (ii) up-regulation by NaCl shock (0.3 M NaCl for 10 min) (e.g. *proVW*, *ilvX*), (iii) up-regulation during late exponential/early stationary phase of growth (e.g. *sopE2*, *sipBCDA*, *siiABCDEF*) (S1 Dataset). This suggests that cytosolic bacteria are growing fast and exposed to a low iron, high osmolarity physicochemical environment. The rapid growth of cytosolic *S*. Typhimurium has been previously observed [12,13,32], validating the former prediction. A common theme observed for the 443 genes up-regulated in the vacuole was induction by: (i) *in vitro* SPI2 growth conditions and/or (ii) internalization into macrophages (S1 Dataset).

## Cytosolic iron-limitation is a cue for *S*. Typhimurium gene induction

To independently assess the cytosolic expression of genes predicted by RNA-seq analysis, we used green fluorescent protein (GFP)-based transcriptional fusions. This fluorophore is an ideal reporter to study differential gene expression in individual bacteria in infected host cells and tissues [12,18,48,49]. Previously *gfpmut3* transcriptional fusions have demonstrated the

vacuole-specific induction of SPI2 genes [50], for example. Due to the low abundance of many transcripts in the "up cytosol" shortlist (according to TPM; S1 Dataset), we also chose to use *gfpmut3*, which encodes for a stable GFP variant, to increase the likelihood of detecting fluorescent bacteria in epithelial cells. We initially focused on those genes that are induced when *Salmonella* are exposed to low $Fe^{2+}$ and/or bile shock in broth [34] (S1 Dataset). Epithelial cells were infected with mCherry-*S*. Typhimurium carrying individual *gfpmut3* reporters and bacterial GFP fluorescence intensity was assessed qualitatively and quantitatively at 8 h p.i. The following genes were up-regulated in the cytosol: *entC* (*entCEBA* operon), *fepA* (*fepA-entD* operon), *fepB*, *fhuA* (*fhuACBD* operon), *fhuE*, *iroB* (*iroBCDE* operon), *iroN*, *mntH*, *nrdH* (*nrdHIEF* operon), *sitA* (*sitABCD* operon), *SL1802*, *SL3990* (*SL3990-SL3989* operon), *sufA* (*sufABCDSE* operon), *yjjZ* (SL1344_4483), STnc3080 and STnc3250 (Figs 3 and S3). Up-regulation of STnc4000, *ilvC* and SL1344_2715 was not observed by fluorescence microscopy (S4 Fig). Quantification using ImageJ software of the mean fluorescence intensity (MFI) of the GFP signal associated with individual bacteria confirmed the qualitative analysis for *iroB* (11-fold increase in average MFI for cytosolic bacteria/vacuolar bacteria, n = 2 experiments), *nrdH* (11-fold), *sitA* (13-fold), *sufA* (7-fold), *yjjZ* (23-fold) and STnc3080 (11-fold) (Fig 3, lower panel). Bacteria harboring these reporters and grown to late log-phase in LB-Miller broth, i.e. $t_0$ conditions, showed only background levels of fluorescence (Fig 3, lower panel).

All of the confirmed *S*. Typhimurium genes are up-regulated by "low Fe2+ shock" in broth (addition of 200 μM DPI for 10 min [34]). DPI chelation is not specific to iron, however [51,52], raising the possibility that other metals also regulate the expression of these genes. We used *iroN*, *sitA*, *yjjZ*, *sufA*, *fepA* and STnc3250 promoters to determine metal cation specificity *in vitro*. The effect of increasing concentrations (0.1–100 μM) of $Co^{2+}$, $Fe^{3+}$, $Mn^{2+}$, $Ni^{2+}$ or $Zn^{2+}$ on GFP fluorescence for bacteria grown in defined minimal media (M9) was measured (S5 Fig). For all reporters, maximal expression was observed in the absence of any added metal. Upon addition of $Fe^{3+}$ and $Co^{2+}$, GFP expression of *S*. Typhimurium carrying P*iroN-gfpmut3*, P*yjjZ-gfpmut3* and P*fepA-gfpmut3* reporters decreased in a dose-dependent manner. Notably, the promoter responses were 10- to 100-fold more sensitive to $Fe^{3+}$ than $Co^{2+}$. P*sitA-gfpmut3* expression was modulated by $Fe^{3+}$, $Mn^{2+}$ and $Co^{2+}$, although $Fe^{3+}$ and $Mn^{2+}$ were clearly the most potent repressors by an order of magnitude, in agreement with a previous study using a *sitA*::*lacZ* reporter [53]. Lastly, expression of P*STnc3250-gfpmut3* and P*sufA-gfpmut3* were only reduced by increased concentrations of $Fe^{3+}$. P*iroN-gfpmut3* was the most sensitive iron reporter, with a >50% reduction in fluorescence at 0.1 μM $Fe^{3+}$. Collectively, the *in vitro* repression of these genes/sRNAs by 0.1–1 μM $Fe^{3+}$ hinted that the induction cue in the cytosol was iron limitation (<0.1 μM).

To test the response of reporters to iron concentrations during infection, we used ferric ammonium citrate (FAC) to increase free iron levels in mammalian cells [54,55]. Epithelial cells were incubated overnight in growth media containing increasing concentrations (10–300 μM) of FAC. Untreated and FAC-treated cells were infected with mCherry-*S*. Typhimurium carrying transcriptional reporters, fixed at 8 h p.i. and the MFI of GFP signal for cytosolic and vacuolar bacteria quantified by fluorescence microscopy and ImageJ (Fig 4). The fluorescence intensity of cytosolic bacteria harboring P*iroN-gfpmut3*, P*sitA-gfpmut3* or P*yjjZ-gfpmut3* reporters was reduced in a dose-dependent manner over the FAC concentration range. Specifically, the average MFI was reduced by 57-, 36- and 15-fold, respectively, upon the addition of 300 μM FAC to growth media (Fig 4A), indicating repression of *iroN*, *sitA* and *yjjZ* expression in the cytosol under iron-replete conditions. The low MFI for vacuolar bacteria in untreated cells limited our analysis of the iron-dependent repression of *iroN*, *sitA* and *yjjZ* in the SCV.

We used the lipophilic, cell-permeable chelator, DPI, as an independent way to assess the effects of metals on *S*. Typhimurium gene expression in the mammalian cytosol. Because DPI

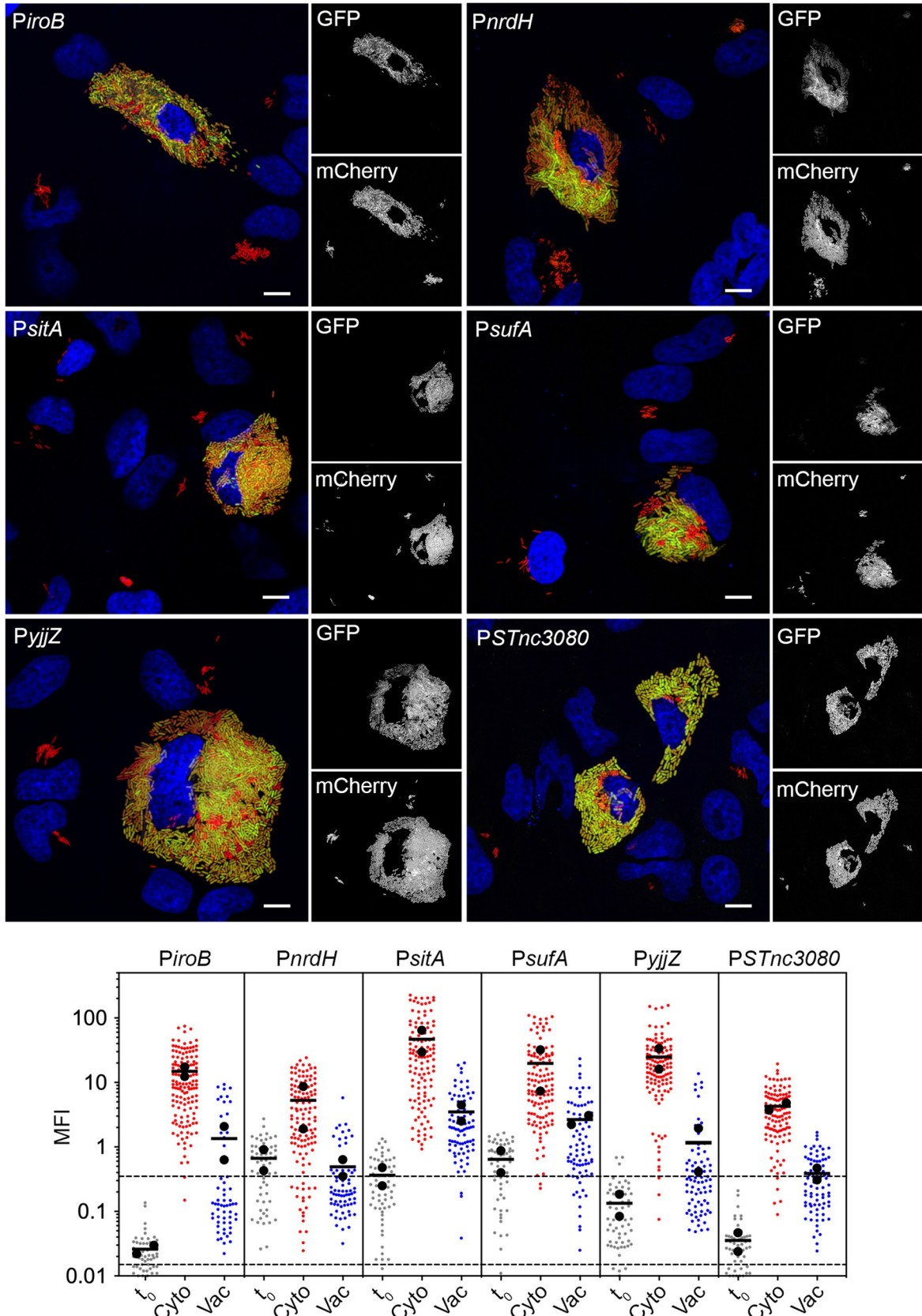

**Fig 3. Iron-associated genes are induced in the epithelial cytosol.** (A) Epithelial cells seeded on coverslips were infected wild-type mCherry-*S.* Typhimurium harboring *gfpmut3* transcriptional reporters. At 8 h p.i., cells were fixed and stained with Hoechst 33342 to detect DNA. Representative confocal microscopy images show induction of *iroB*, *nrdH*, *sitA*, *sufA*, *yjjZ* and STnc3080 promoters in cytosolic bacteria. Green = transcriptional reporter, red = *S.* Typhimurium, blue = DNA. Scale bars are 10 μm. (B) Quantification of the mean fluorescence intensity (MFI) of GFP signal by fluorescence microscopy and ImageJ. Bacteria were designated as being cytosolic (Cyto) or vacuolar (Vac) if residing within cells with ≥100 bacteria or 2–40 bacteria, respectively. $t_0$ represents the infection inoculum i.e. bacteria grown to late log phase in LB-Miller broth. Small dots represent individual bacteria; large dots indicate the mean of each experiment; horizontal bars indicate the average of two experiments. Acquisition parameters (exposure time and gain) were set up using P*sitA-gfpmut3* (the highest GFP intensity) and these same parameters were applied throughout. Dashed lines indicate the range of background fluorescence in the GFP channel measured for mCherry-*S.* Typhimurium (no reporter plasmid).

has a high affinity for various metal cations, we assessed the specificity for iron by means of an "add-back" experimental design [51,56,57] whereby an equimolar amount of $Fe^{3+}$ was added to the DPI-treated media. Epithelial cells were treated overnight with 200 μM DPI, 200 μM DPI plus 200 μM FAC, or 200 μM FAC alone and infected with mCherry-*S.* Typhimurium harboring P*iroN-gfpmut3*, P*sitA-gfpmut3* or P*yjjZ-gfpmut3* reporters. Cells were fixed at 8 h p.i. and the intensity of GFP signal for cytosolic and vacuolar bacteria quantified via fluorescence microscopy. Addition of DPI increased the expression of *iroN*, *sitA* and *yjjZ* for both cytosolic and vacuolar bacteria (Fig 4B), indicating that DPI accesses both cellular compartments. However, the effects of divalent cation chelation were more pronounced for the vacuolar population (13- to 43-fold increase in average MFI) than the cytosolic population (1.7- to 2.2-fold increase in average MFI). Add-back of iron restored the MFI to that of untreated cells for cytosolic bacteria harboring P*iroN-gfpmut3* and P*sitA-gfpmut3* reporters (Fig 4B), demonstrating that iron limitation is the primary stimulus for the up-regulation of these genes in the cytosol. Add-back of iron had a lesser effect on the average MFI of bacteria harboring the P*yjjZ-gfpmut3* reporter, regardless of their intracellular niche (Fig 4B), possibly due to a lower sensitivity of *yjjZ* to changes in free iron levels (Fig 4A) or a responsiveness to deprivation of metals other than iron in the intracellular environment (S5 Fig). Add-back of iron only partially restored the expression of *sitA* and *iroN* in vacuolar bacteria (Fig 4B) suggesting that *sitA* and *iroN* respond to multiple divalent cations in the SCV or FAC treatment is less effective at increasing free iron levels in endocytic-derived compartments compared to the cytosol. Taken together, our data show that iron limitation in the mammalian cytosol is a major cue sensed by *S.* Typhimurium.

Using the same treatment conditions, we determined the role of iron limitation on bacterial replication in epithelial cells. As assessed by gentamicin protection assay (Fig 4C), total bacterial replication was limited by treatment with DPI (4.0-fold replication over 8 h) and DPI plus FAC (5.6-fold), but not FAC alone (8.8-fold), compared to untreated cells (10.0-fold). The CHQ resistance assay showed a similar profile (Fig 4D). Compared to untreated cells (50 ±3.4% cytosolic bacteria), treatment with DPI reduced the levels of cytosolic bacteria to 28 ±6.7% at 7 h p.i. The proportion of cytosolic bacteria was increased to 42±11% upon add-back of iron (DPI+FAC), whereas it was indistinguishable from untreated cells for FAC alone (52 ±5.5%). Notably, DPI treatment (200 μM) also retards the growth of *S.* Typhimurium in complex media (S6 Fig).

It has been shown that epithelial cells contain either cytosolic bacteria, vacuolar bacteria or a mix of both populations at later times p.i. [32]. We used single-cell analysis to monitor the effect of metals on the distribution of these intracellular populations. Using mCherry-*S.* Typhimurium harboring pFN101 as a fluorescent reporter for cytosolic access, we analyzed the intracellular populations at 8 h p.i., in addition to the number of cytosolic (GFP+, mCherry+) and vacuolar (GFP-, mCherry+) bacteria within each infected cell. There was no significant difference in the frequency of epithelial cells that only contained cytosolic bacteria (0–1.2% of

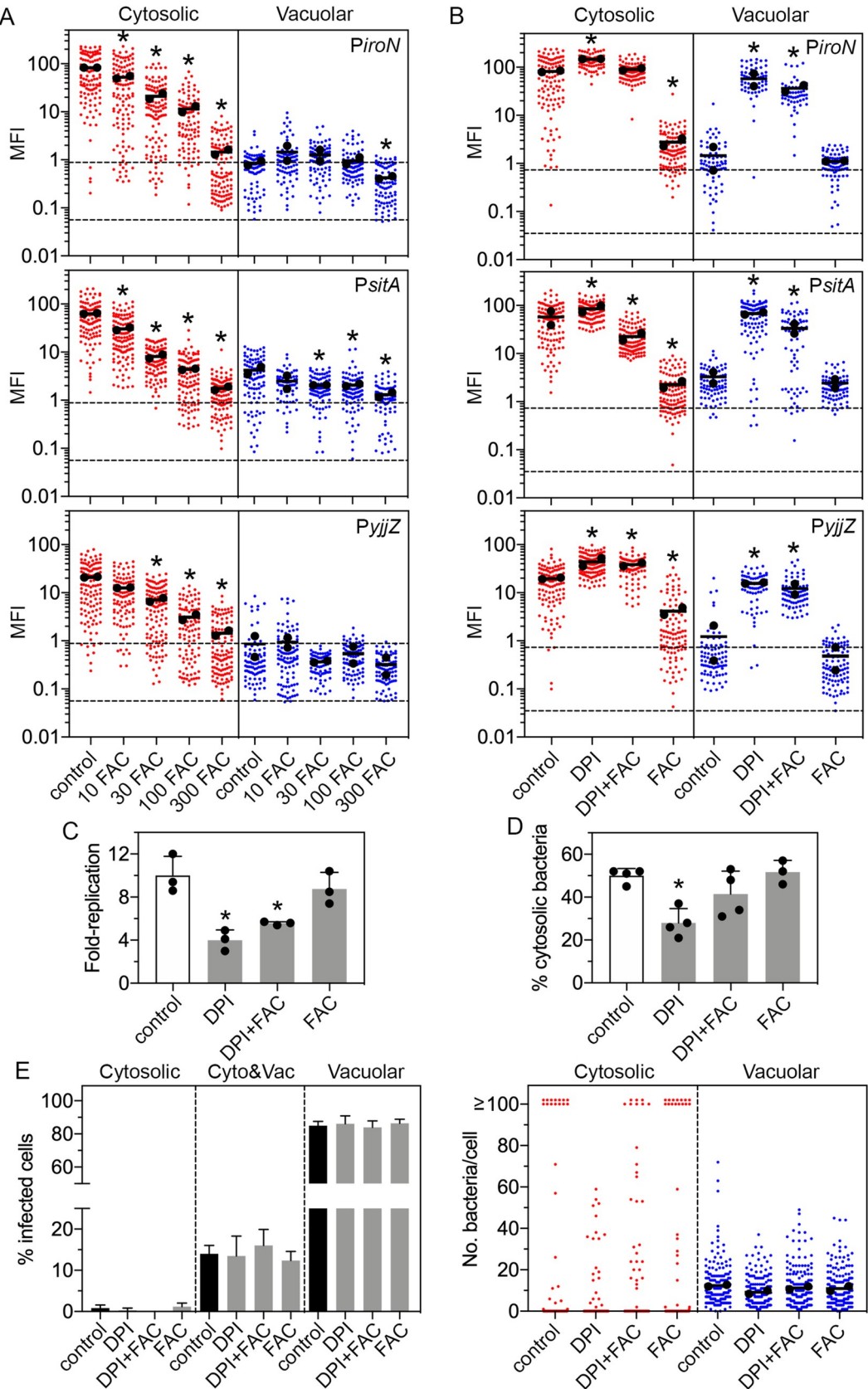

**Fig 4. Manipulation of cellular iron levels primarily affects cytosolic bacteria.** (A) Epithelial cells seeded on coverslips were left untreated (control) or treated overnight in growth media supplemented with 10 µM, 30 µM, 100 µM or 300 µM ferric ammonium citrate (FAC). Cells were infected with wild-type mCherry-*S*. Typhimurium harboring P*iroN-gfpmut3*, P*sitA-gfpmut3* or P*yjjZ-gfpmut3* reporters, fixed at 8 h p.i. and the MFI of bacterial GFP signal was quantified by fluorescence microscopy and Image J. Bacteria were designated as being cytosolic (Cyto) or vacuolar (Vac) if residing within cells with ≥100 bacteria or 2–40 bacteria, respectively. Small dots represent individual bacteria; large dots indicate the mean of each experiment; horizontal bars indicate the average of two experiments. Acquisition parameters (exposure time and gain) were set-up using P*iroN-gfpmut3* (the highest GFP intensity) and these same parameters were applied throughout. Dashed lines indicate the range of background fluorescence in the GFP channel measured for mCherry-*S*. Typhimurium (no reporter plasmid). Asterisks indicate data significantly different from control (Kruskal-Wallis test). (B) Epithelial cells seeded on coverslips were left untreated (control) or treated overnight with growth media containing 200 µM 2,2'-dipyridyl (DPI), a metal-chelating compound, 200 µM DPI and 200 µM ferric ammonium citrate (DPI+FAC), or 200 µM FAC alone (FAC). Infection strains and quantification of GFP signal were as described for (A). Asterisks indicate data significantly different from control (Kruskal-Wallis test). (C) Cells were treated as in (B) and infected with wild-type bacteria. The number of intracellular bacteria was quantified by gentamicin protection assay at 1 h and 8 h p.i. Fold-replication is CFUs at 8 h/1 h. The mean from each experiment is represented as a large dot (n = 3). Asterisks indicate data significantly different from control (ANOVA with Dunnett's post-hoc test). (D) Cells were treated as in (B) and infected with wild-type bacteria. The proportion of cytosolic bacteria was quantified by CHQ resistance assay at 7 h p.i. The mean from each experiment is represented as a large dot (n≥3). Asterisk indicates data significantly different from control (ANOVA with Dunnett's post-hoc test). (E) Cells were treated as in (B) and infected with mCherry-*S*. Typhimurium harboring a plasmid-borne reporter of cytosolic access, pNF101, and fixed at 8 h p.i. Left panel: the proportion of infected cells containing only cytosolic (all bacteria are GFP+, mCherry+), only vacuolar (all bacteria are GFP-, mCherry+) or a mixed population (Cyto&Vac) of bacteria was blindly scored by fluorescence microscopy. n≥5 independent experiments. Right panel: the number of cytosolic or vacuolar bacteria in each cell was blindly scored. Small dots represent individual bacteria; large dots indicate the mean of each experiment; horizontal bars indicate the average of two experiments.

infected cells), only vacuolar bacteria (84.0–86.4% of infected cells) or a mixed population (12.4–16.0% of infected cells) between any of the treatments (Fig 4E, left panel), indicating that metal chelation does not alter the intracellular distribution of bacteria. However, the ability of *Salmonella* to replicate in the cytosol was noticeably affected (Fig 4E, right panel), in agreement with the CHQ resistance assay results. In untreated (control) cells, 6.4±0.89% of infected cells contained ≥100 cytosolic bacteria/cell at 8 h p.i. (n = 4 experiments). DPI treatment reduced proliferation of *S*. Typhimurium in the cytosol such that only 0.40±0.89% of infected cells contained ≥100 cytosolic bacteria/cell. Add-back of iron (DPI+FAC) partially restored the hyper-proliferative capacity of *S*. Typhimurium in the cytosol (3.3±1.9%) whereas FAC treatment alone (7.8±1.7%) was no different than control. DPI limited bacterial replication in the SCV (Fig 4E, right panel) as evidenced by a reduction in the average number of vacuolar bacteria/cell at 8 h p.i. (13.8 for untreated and 10.8 for DPI-treated, n = 4 experiments), albeit without statistical significance (p = 0.15, Kruskal-Wallis test). Add-back of iron restored vacuolar replication (11.7 bacteria/cell) whereas FAC alone (11.6 bacteria/cell) was not significantly different to untreated cells. In summary, iron restriction affects *S*. Typhimurium proliferation in both the cytosol and vacuole in epithelial cells but has a much more profound effect in the cytosol.

## Zinc and magnesium are more limiting in the vacuole than the cytosol

We investigated whether metals other than iron affected bacterial gene expression in epithelial cells. *S*. Typhimurium encodes multiple transporters for divalent cations [58]. The up-regulation of *sitABCD* and *mntH* in cytosolic *S*. Typhimurium (Figs 3 and S3), which are induced by iron and manganese limitation (S5 Fig) [53,56] and encode high affinity $Mn^{2+}$ transporters [59,60], suggests that $Mn^{2+}$ availability is lower in the cytosol than SCV. $Zn^{2+}$ import is dependent on a high affinity transport system in *S*. Typhimurium, ZnuABC [61], and its accessory protein, ZinT (formerly known as YodA) [62,63], and a low affinity ZIP family metal permease, ZupT [64,65]. *zupT* is constitutively expressed [64], in agreement with our RNA-seq analysis (S1 Dataset). By contrast, *znuA* and *zinT* are induced when zinc is limiting [66] and both

genes are in the "up vacuole" shortlist (Fig 2A and S1 Dataset), consistent with increased ZnuA production upon *S.* Typhimurium internalization into mammalian cells [61]. We found that a P*zinT-gfpmut3* reporter was exquisitely sensitive to zinc concentrations, with a 54% and 99% reduction in GFP fluorescence detected upon the addition of 0.1 μM and 1 μM $Zn^{2+}$ to minimal media, respectively (S5 Fig). The fluorescence intensity of both vacuolar and cytosolic mCherry-*S.* Typhimurium harboring P*zinT-gfpmut3* increased at 8 h p.i. compared to $t_0$, indicating the intracellular induction of *zinT*. However, GFP signal was considerably greater for vacuolar bacteria (Fig 5), suggesting that zinc is more limiting in the SCV (<0.1 μM). Quantification of the average MFI of GFP fluorescence confirmed the qualitative findings (5.8-fold increase for vacuolar bacteria/cytosolic bacteria; Fig 5, lower panel). A recent study utilizing a *znuA* transcriptional reporter to measure intracellular $Zn^{2+}$ levels reached a similar conclusion [67].

*S.* Typhimurium also has multiple transporters for $Mg^{2+}$ uptake. CorA is a constitutively expressed $Mg^{2+}$ transporter, and MgtA and MtgB are inducible $Mg^{2+}$ transporters [68]. Consistent with its constitutive expression, expression of *corA* was similar in both vacuolar and cytosolic bacteria (S1 Dataset). Low $Mg^{2+}$ induces *mgtA* and *mgtB* transcription [68,69]. Expression of *mgtA* was not induced in the vacuole, however (S1 Dataset). Acidic pH abolishes the *mgtA* (but not *mgtB*) transcriptional response to $Mg^{2+}$ concentration [69], which could explain why *mgtA* is not induced in the SCV of epithelial cells. In contrast, all three genes within the *mgtCBR* operon were up-regulated in vacuolar bacteria according to the RNA-seq dataset (S1 Dataset). Using mCherry-*S.* Typhimurium P*mgtC-gfpmut3*, we confirmed *mgtCBR* is responsive to $Mg^{2+}$ levels (S5 Fig) and the operon is up-regulated after bacterial internalization (Fig 5). Furthermore, the average MFI of GFP signal was increased by 6.2-fold for vacuolar bacteria/cytosolic bacteria (Fig 5, lower panel) indicating that $Mg^{2+}$ is more limiting in the SCV than cytosol, in agreement with a recent study [67]. Altogether, our studies of metal-regulated gene expression reveal that *S.* Typhimurium encounters distinct metal ion availabilities in the SCV and cytosol; specifically, iron and manganese are more limiting in the cytosol whereas magnesium and zinc are limiting in the vacuole.

## Identification of growth-phase related, cytosol-induced genes

Our transcriptomic analysis highlighted another set of cytosol-induced genes that are highly expressed during late exponential/early stationary phase of *in vitro* growth in broth, and in some cases responsive to oxygen shock (Fig 2 and S1 Dataset). Many of the genes encode for T3SS1 structural proteins (*orgA*, *spaQ*, *sipB*, *sipC*, *sipD*) and effectors known to be translocated by T3SS1 (*sipA*, *sopB*, *sopD*, *sopE*, *sopE2*, *sopF* [SL1344_1177]). We have previously shown that *prgH* (*prgHIJK-orgABC* operon), encoding a structural T3SS1 component, is up-regulated in cytosolic bacteria [12], in agreement with our RNA-seq results (S1 Dataset). Using GFP transcriptional reporters, we confirmed the increased expression of *sicA* (*sicA-sipBCDA* operon), *sopE2* and *sopF* (Fig 6) in cytosolic bacteria. Of note, the average MFI of cytosolic bacteria was comparable to $t_0$ cultures (Fig 6, lower panel), which are highly T3SS1-induced [70]. PipB, a T3SS2 effector encoded on SPI5 [71], was used as a control for the vacuole-specific induction of a type III effector. In accordance, the average MFI of GFP fluorescence was 9.8-fold higher for vacuolar bacteria harboring a P*pipB-gfpmut3* reporter compared to those in the cytosol (Fig 6). We confirmed that *siiA* (*siiABCDEF* operon) and *lpxR* were more highly expressed in bacteria residing within the cytosol compared to the SCV (Fig 6). The *siiABCDEF* operon in SPI4 (S2 Fig) encodes for a type I secretion system for the secretion of SiiE, a giant non-fimbrial adhesin [72,73]. Immunostaining with anti-SiiE antibodies detected the adhesin on the surface of cytosolic bacteria (S7 Fig, upper panel), and diffusely spread throughout the cytosol

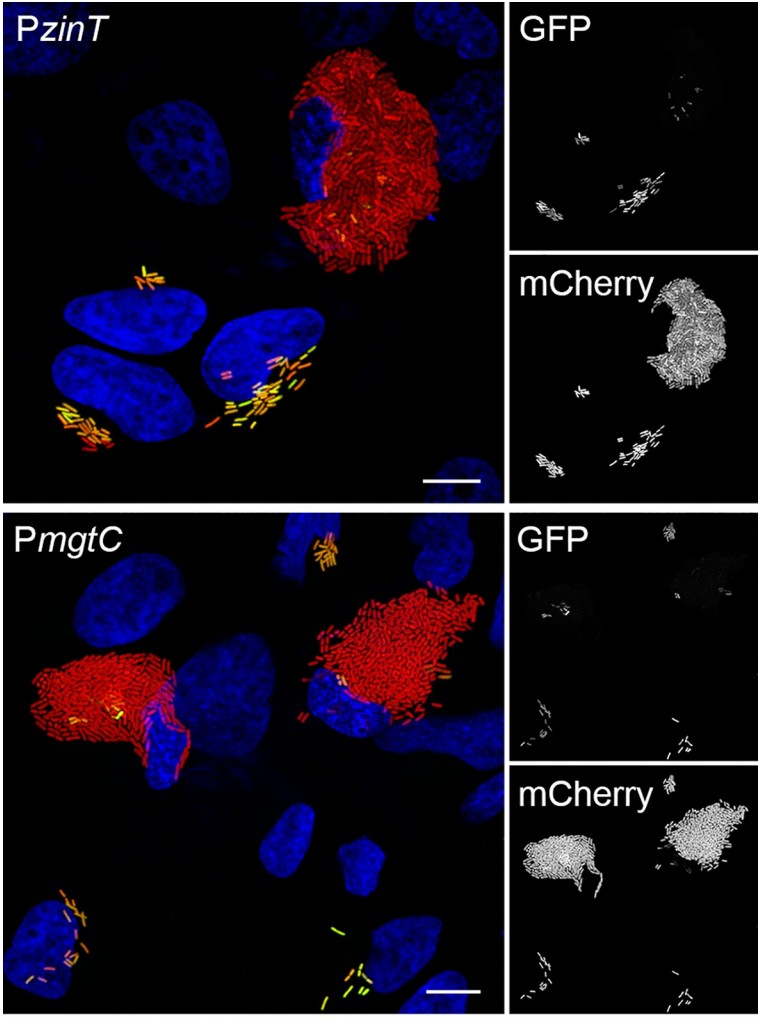

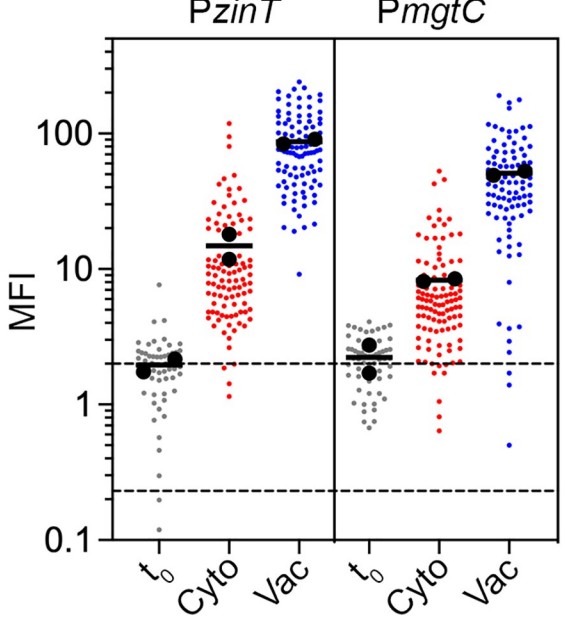

**Fig 5. Zn$^{2+}$ and Mg$^{2+}$ are limiting in the vacuole, not the cytosol.** Upper panels: Epithelial cells seeded on coverslips were infected with mCherry-*S*. Typhimurium harboring P*zinT-gfpmut3* or P*mgtC-gfpmut3* transcriptional reporters. At 8 h p.i., cells were fixed and stained with Hoechst 33342 to detect DNA. Representative confocal microscopy images show induction of *zinT* (upper panel) *and mgtC* (lower panel) promoters in vacuolar bacteria. Green = transcriptional reporter, red = *S*. Typhimurium, blue = DNA. Scale bars are 10 μm. Lower panels: Quantification of the MFI of GFP signal by fluorescence microscopy and ImageJ. Bacteria were designated as being cytosolic (Cyto) or vacuolar (Vac) if residing within cells with ≥100 bacteria or 2–40 bacteria, respectively. t$_0$ represents the infection inoculum i.e. bacteria grown to late log phase in LB-Miller broth. Small dots represent individual bacteria; large dots indicate the mean of each experiment; horizontal bars indicate the average of two experiments. Acquisition parameters (exposure time and gain) were set up using P*zinT-gfpmut3* (the highest GFP intensity) and these same parameters were applied throughout. Dashed lines indicate the range of background fluorescence in the GFP channel measured for mCherry-*S*. Typhimurium (no reporter plasmid).

(S7 Fig, lower panel). Vacuolar bacteria were negative for SiiE staining (S7 Fig). We have previously shown by electron microscopy that cytosolic bacteria have extensive filamentous material on their surface [12]. Taken together, our findings suggest the filaments could be SiiE. Like the *sii* operon, *lpxR* (SL1344_1263) is also co-regulated with SPI1 [74] and encodes an outer membrane 3'-*O*-deacylase of lipid A [75,76]. Transcriptional reporters did not verify cytosolic up-regulation of *ygbA* (SL1344_2840), which encodes a conserved hypothetical protein required for *S*. Typhimurium growth during nitrosative stress [77], or *asnA*, which is involved in asparagine biosynthesis [78] (S4 Fig).

We conclude that there is a generalized up-regulation of "SPI1 associated" genes in the mammalian cytosol. However, we do not observe uniform up-regulation of these genes within the cytosolic population (Fig 6), in agreement with previous reports for *prgH* [12,14,36]. Based on average MFIs, all genes were more highly expressed in cytosolic bacteria than vacuolar bacteria (Fig 6, lower panel)–*sicA* (2.4-fold), *sopE2* (6.1-fold), *sopF* (3.8-fold), *siiA* (8.5-fold) and *lpxR* (2.3-fold)–yet with a broader range of MFI/bacterium than the iron-regulated genes (Fig 3, lower panel), and with many cytosolic bacteria having background levels of GFP fluorescence. Such heterogeneity in T3SS1 gene expression has been reported previously in broth cultures [79,80], a result we confirmed here (t$_0$; Fig 6, lower panel).

## Extraneous genes that are induced in the epithelial cytosol

Several osmotically-sensitive genes that are up-regulated when *S*. Typhimurium is exposed to NaCl shock [34] were identified as candidate cytosol-induced genes, namely *cysP* (*cysPUWA* operon), *soxS* and *proV* (*proVWX* operon) (S1 Dataset). We qualitatively and quantitatively confirmed the cytosol-specific expression of *cysP* and *soxS* using transcriptional reporters (S8 Fig). However, the *proV-gfpmut3* transcriptional fusion was not up-regulated in the cytosol (S4 Fig). The *cysPUWA* operon encodes a sulfate/thiosulfate permease and periplasmic binding protein [81]. SoxS is a member of the AraC/XylS family of transcriptional regulators that is required for bacterial resistance to oxidative stress [82,83].

Six additional *S*. Typhimurium genes were identified as being up-regulated in the cytosol according to the transcriptomics data, namely *uhpT*, *sfbA* (*sfbABC* operon), *fruB* (*fruBKA* operon), *grxA*, *mtr* and *trpE* (*trpEDCBA* operon) (S1 Dataset). All six genes are constitutively expressed in infection-relevant broth conditions [34] and *sfbA*, *mtr*, *trpE* and *fruB* are also up-regulated upon *S*. Typhimurium infection of macrophages ([43]; S1 Dataset). Infection of epithelial cells with mCherry-*S*. Typhimurium harboring P*uhpT-gfpmut3*, P*sfbA-gfpmut3*, P*grxA-gfpmt3* and P*fruB-gfpmut3* confirmed the RNA-seq-based prediction of their induction in the cytosol (S8 Fig). However, *mtr* and *trpE* were not induced in the cytosol (S4 Fig). UhpT is a hexose phosphate transporter whose induction has been reported previously for cytosolic *S*. Typhimurium [36]. *fruBKA*, known as the fructose operon, encodes for three enzymes

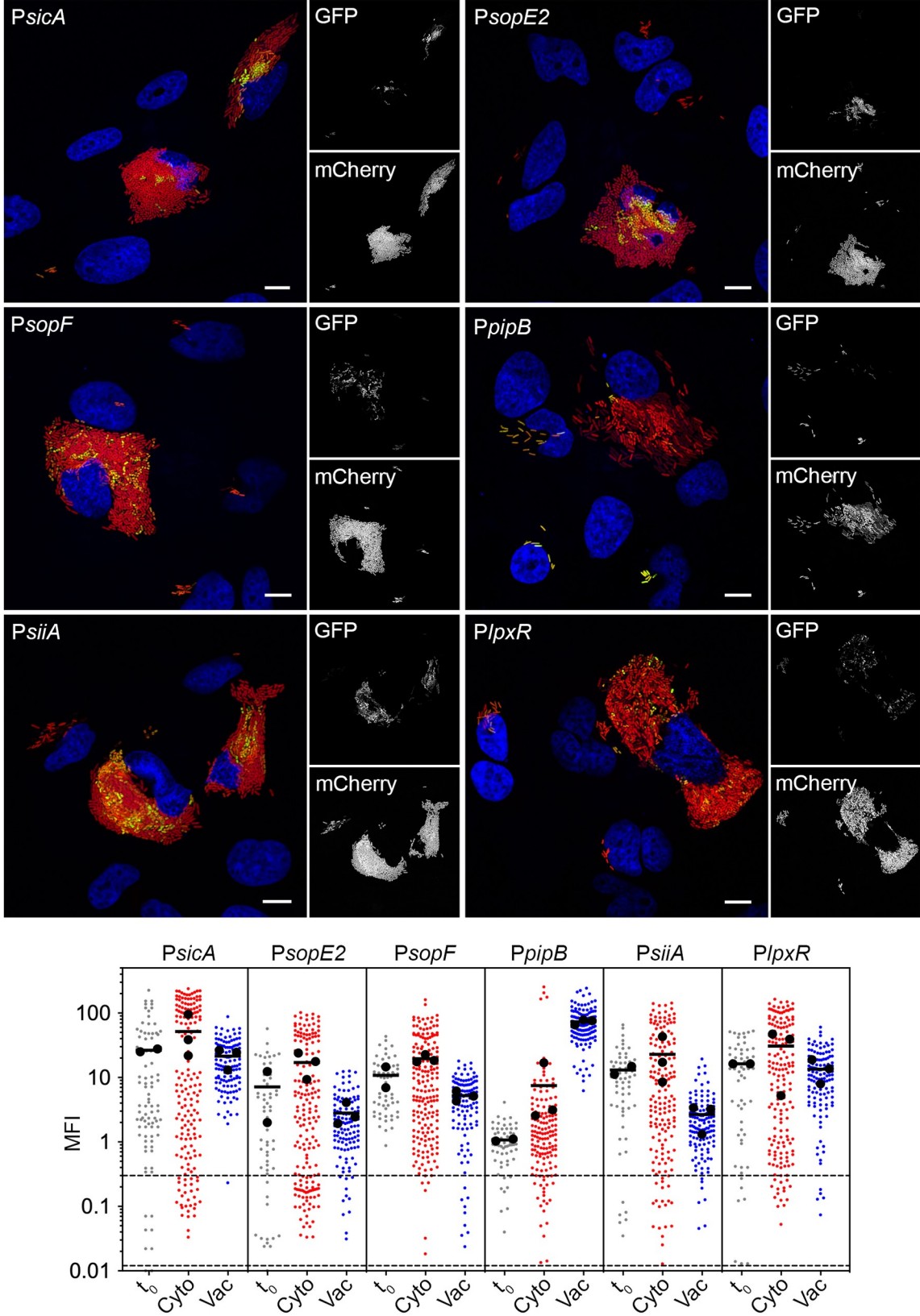

**Fig 6. SPI1-associated genes are up-regulated in the epithelial cytosol.** (A) Epithelial cells seeded on coverslips were infected with mCherry-*S*. Typhimurium harboring *gfpmut3* transcriptional reporters. At 8 h p.i., cells were fixed and stained with Hoechst 33342 to detect DNA. Representative confocal microscopy images show induction of *sicA*, *sopE2*, *sopF*, *siiA* and *lpxR* promoters in cytosolic bacteria. PipB is a type III effector translocated by T3SS2 and the P*pipB-gfpmut3* reporter served as a control for vacuole-specific gene induction. Green = transcriptional reporter, red = *S*. Typhimurium, blue = DNA. Scale bars are 10 μm. (B) Quantification of the MFI of GFP signal by fluorescence microscopy and ImageJ. Bacteria were designated as being cytosolic (Cyto) or vacuolar (Vac) if residing within cells with ≥100 bacteria or 2–40 bacteria, respectively. $t_0$ represents the infection inoculum i.e. bacteria grown to late log phase in LB-Miller broth. Small dots represent individual bacteria; large dots indicate the mean of each experiment; horizontal bars indicate the average of 2–3 experiments. Acquisition parameters (exposure time and gain) were set up using P*sicA-gfpmut3* (the highest GFP intensity) and these same parameters were applied throughout. Dashed lines indicate the range of background fluorescence in the GFP channel measured for mCherry-*S*. Typhimurium (no reporter plasmid).

involved in fructose uptake [84]. GrxA, or glutaredoxin 1 (Grx1), is a redox enzyme that detoxifies oxidizing agents such as reactive oxygen species (ROS), thereby defending against oxidative stress [85]. The *sfbABC* operon is predicted to encode a periplasmic iron-binding lipoprotein SfbA, a nucleotide-binding ATPase SfbB and a cytoplasmic permease SfbC [86].

## Identification of *S*. Typhimurium genes required for optimal cytosolic proliferation

Having identified *S*. Typhimurium genes/sRNAs that are up-regulated within the epithelial cytosol, we next tested whether any of these genes/sRNAs are required for colonization of this niche. We did not focus on the SPI1-associated effectors as their individual contributions to the cytosolic stage of the intracellular cycle have been reported previously [24,32,87,88]. Twenty-nine single or multiple gene deletion mutants were constructed and their ability to access and replicate within the cytosol was first assessed in a population-based assay by CHQ resistance (Fig 7A). Four single gene deletion mutants had a significantly reduced proportion of cytosolic bacteria at 7 h p.i. compared to wild-type bacteria (50.2±7.2%), namely Δ*entC* (41.9±7.0%), Δ*fepB* (40.6±5.9%), Δ*soxS* (42.4±6.4%) and Δ*znuA* (26.3±6.7%). A Δ*soxS*Δ*grxA* mutant (40.0±4.2%) was also defective, but not more than the Δ*soxS* deletion mutant. Deletion of both Mn$^{2+}$ transporters (Δ*mntH*Δ*sitA*) reduced the proportion of cytosolic bacteria (40.1 ±6.2%), whereas single Δ*mntH* and Δ*sitA* deletion mutants were without effect. Lastly, a triple Δ*mntH*Δ*sitA*Δ*entC* mutant was more compromised (29.8±4.3%) for the percentage of cytosolic bacteria at 7 h p.i. than the Δ*mntH*Δ*sitA* mutant or Δ*entC* mutant. None of the identified mutants had a growth defect in complex media (S6 Fig). The remaining deletion mutants had similar proportions of cytosolic bacteria as wild-type *S*. Typhimurium.

Plasmid-borne complementation of Δ*entC* (with *entCEBA*), Δ*grxA*Δ*soxS* (with *soxS*), Δ*mntH*Δ*sitA* (with *sitABCD*) and Δ*znuA* (with *znuA*) mutants restored their respective phenotypes to wild-type levels in the CHQ resistance assay (Fig 7B). Chromosomal complementation by Tn7-based integration at the *glmS* site with *fepB* also restored the Δ*fepB* defect to wild-type levels (Fig 7B). Importantly, Δ*entC*, Δ*fepB*, Δ*grxA*Δ*soxS*, Δ*mntH*Δ*sitA*, Δ*mntH*Δ*sitA*Δ*entC*, Δ*soxS* and Δ*znuA* mutants were comparable to wild-type bacteria for their proportion of cytosolic bacteria at 90 min p.i. (Fig 7C), indicating that the observed defects at 7 h p.i. are not simply due to delayed vacuole lysis.

We extended this population-based assay with single-cell analysis using *S*. Typhimurium harboring the dual fluorescence reporter plasmid, pCHAR-Duo(ASV), in which constitutive expression of *mCherry* is driven by the synthetic promoter ProB and destabilized GFP (*gfpmut3.1*(ASV)) is under the control of the *S*. Typhimurium *uhpT* promoter. All bacteria harboring this plasmid are mCherry-positive and only those bacteria in the cytosol will be GFP-positive. Epithelial cells were infected with wild-type, Δ*entC*, Δ*fepB*, Δ*mntH*Δ*sitA*, Δ*mntH*Δ*sitA*Δ*entC*, Δ*soxS* and Δ*znuA* bacteria harboring pCHAR-Duo(ASV) and the number

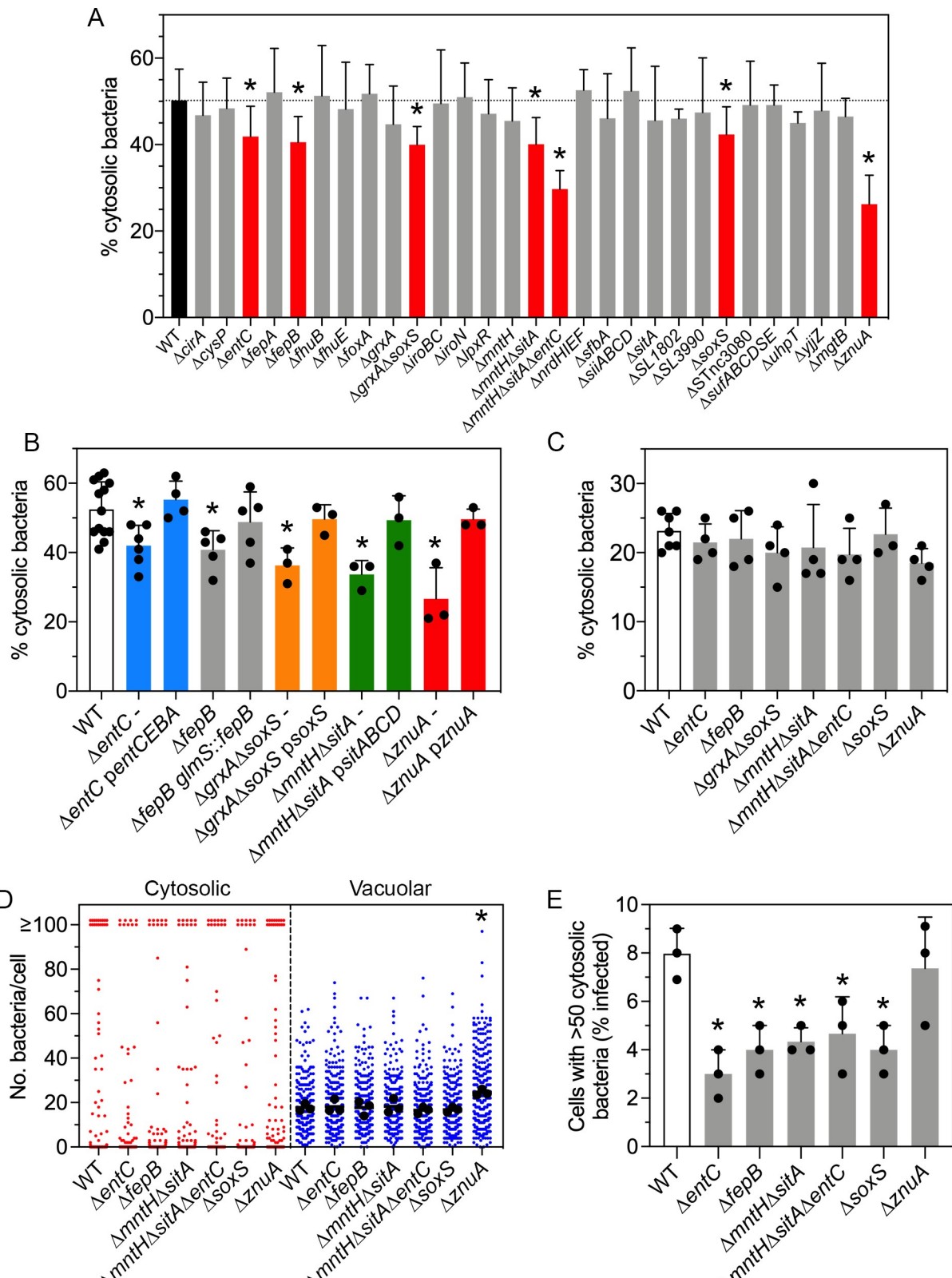

**Fig 7. A subset of "up cytosol" genes is required for optimal proliferation in the epithelial cell cytosol.** (A) Epithelial cells were infected with wild-type (WT) bacteria or the indicated gene deletion mutants and the proportion of cytosolic bacteria at 7 h p.i. was quantified using

the CHQ resistance assay. n≥4 independent experiments for each strain. Asterisks indicate data significantly different from WT (ANOVA with Dunnett's post-hoc test). (B) Genetic complementation. Epithelial cells were infected with WT bacteria or deletion mutants harboring an empty vector (-), pWSK29 or pWKS30, or the respective vector encoding the indicated gene(s), except for the Δ*fepB* mutant, which was complemented with a chromosomal copy of *fepB* at the *glmS* site. The proportion of cytosolic bacteria at 7 h p.i. was quantified using the CHQ resistance assay. Large dots indicate the mean of each experiment (n≥3). Asterisks indicate data significantly different from WT (ANOVA with Dunnett's post-hoc test). (C) Epithelial cells were infected with WT bacteria or the indicated gene deletion mutants and the proportion of cytosolic bacteria at 90 min p.i. was quantified using the CHQ resistance assay. Large dots indicate the mean of each experiment (n≥3). (D) Epithelial cells seeded on coverslips were infected with WT bacteria or gene deletion mutants harboring pCHAR-Duo(ASV), a plasmid-borne dual reporter–constitutive *mCherry* expression is driven by the synthetic ProB promoter and *gfpmut3.1*(ASV) (encoding for destabilized GFP) is under the control of the glucose-6-phosphate responsive *uhpT* promoter from *S*. Typhimurium. The number of cytosolic (GFP+, mCherry+) and vacuolar (GFP-, mCherry+) bacteria in each infected cell was blindly scored by fluorescence microscopy. Small dots represent individual bacteria; large dots indicate the mean of each experiment; horizontal bars indicate the average of 3 experiments. Asterisk indicates data significantly different from WT (Kruskal-Wallis test). (E) Data from (D) was reanalyzed to determine the percentage of infected cells containing >50 cytosolic bacteria. Large dots indicate the mean of each experiment. Asterisks indicate data significantly different from WT (ANOVA with Dunnett's post-hoc test).

of cytosolic (GFP+, mCherry+) and vacuolar (GFP-, mCherry+) bacteria in each infected cell at 8 h p.i. was scored by fluorescence microscopy. There was no statistical difference in the proportion of infected cells that contained cytosolic bacteria between wild-type bacteria and the tested deletion mutants–wild-type (14.7±2.5%, n = 3 experiments), Δ*entC* (10.7±2.5%), Δ*fepB* (11.3±4.6%), Δ*mntH*Δ*sitA* (12.3±3.1%), Δ*mntH*Δ*sitA*Δ*entC* (9.3±4.5%), Δ*soxS* (9.0 ±2.0%) and Δ*znuA* (16.7±2.1%) (Fig 7D). However, Δ*entC*, Δ*fepB*, Δ*mntH*Δ*sitA*, Δ*mntH*Δ*si-tA*Δ*entC* and Δ*soxS* mutants were all defective for proliferation in the cytosol compared to wild-type bacteria as demonstrated by a decreased number of infected cells containing >50 cytosolic bacteria at 8 h p.i. (Fig 7E). The decreased proportion of cytosolic bacteria for the Δ*znuA* mutant (Fig 7A) was explained by an increased number of vacuolar bacteria/cell at 8 h p.i. (Fig 7E) rather than a defect in cytosolic proliferation *per se* (Fig 7D).

Overall, our data show that of the 27 bacterial genes we identified as being up-regulated in the epithelial cytosol (Figs 3, 6, S3, and S8), five are specifically required for the optimal proliferation of *S*. Typhimurium in this intracellular niche—*entC*, *fepB*, *mntH*, *sitA* and *soxS*.

## Discussion

Compared to an endocytic-derived vacuole, the host cell cytosol could naïvely be viewed as a nutrient-rich milieu that allows for the efficient growth of bacteria. However, not all pathogens can survive within this niche, and *S*. Typhimurium can only replicate in the cytosol of epithelial cells and certain embryonic fibroblast lines [12,14,21]. Our finding that vastly different virulence gene programs are activated when *S*. Typhimurium colonizes the SCV and cytosol (Fig 8) confirm previous suggestions that the mammalian cytosol is a complex environment that requires pathogen-specific adaptations for efficient bacterial survival and replication [89].

Here we report the "cytosol transcriptional signature" of bacteria by integrating our data with the available gene expression profiles of Gram-negative pathogens that colonize the cytosol (Table 1). We focus on the prototypical cytosolic pathogen, *S. flexneri* [90–93], and uropathogenic *Escherichia coli* (UPEC) [94,95], which forms intracellular bacterial communities (IBCs) in the cytosol of bladder epithelial cells. The most striking feature that emerges is the enrichment of genes that mediate the acquisition of iron (Table 1). *Salmonella* acquires ferric (Fe$^{3+}$) iron by secreting two siderophores, enterobactin and salmochelin, a C-glucosylated form of enterobactin [58,96,97]. Iron-loaded siderophores then transit back into bacteria by first binding the outer membrane receptors, FepA and IroN [98,99], then FepB in the periplasm, and finally across the cytoplasmic membrane via the ABC-type transporter, FepDGC [100]. FhuE, FhuA and FoxA also function as outer membrane receptors to exploit the utilization of ferric siderophores produced by other bacteria [101]. All of the siderophore synthesis

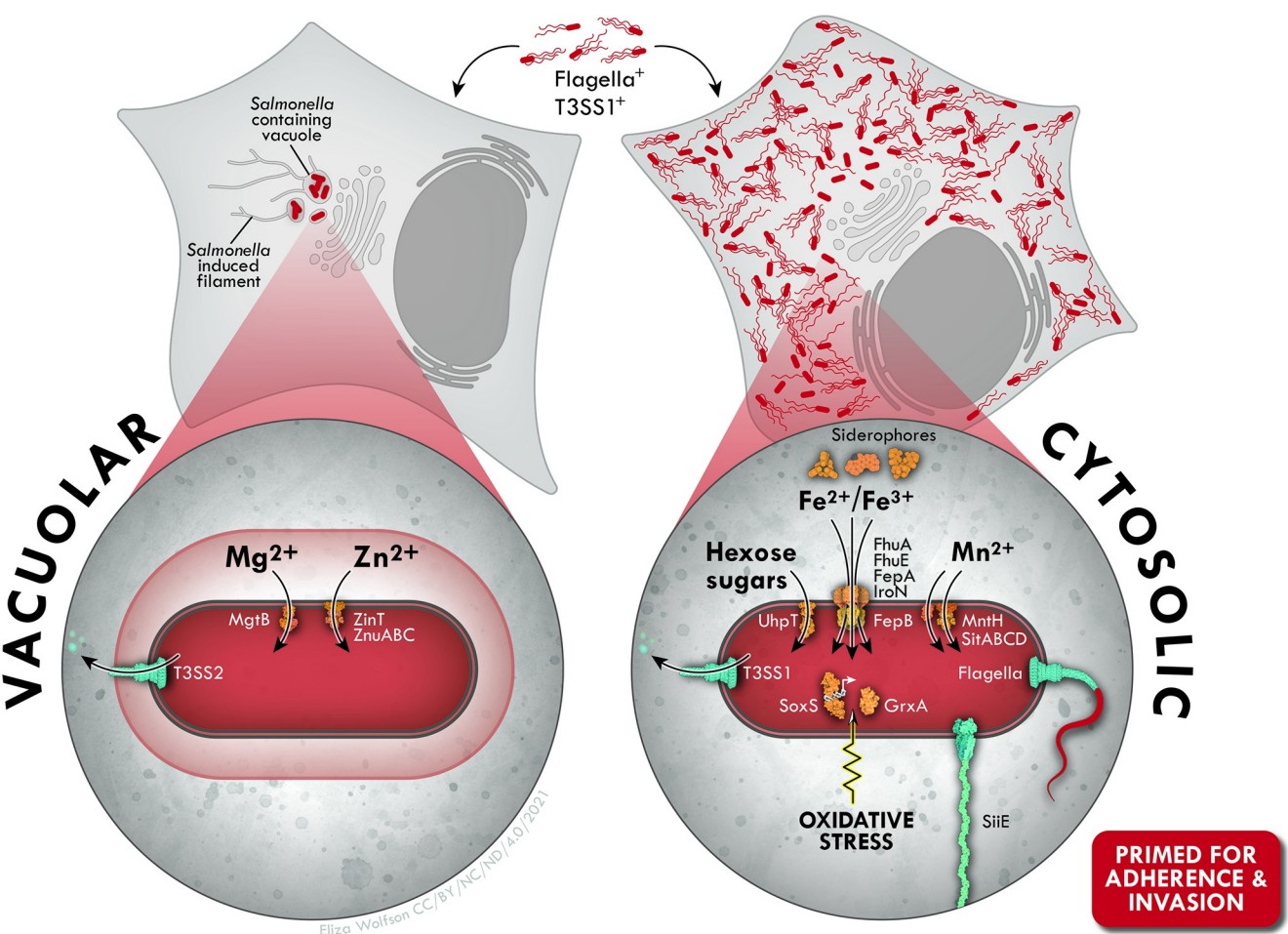

**Fig 8. Adaptation to the epithelial cytosol requires extensive transcriptional reprogramming by S. Typhimurium.** SCV-resident bacteria are translocating type III effectors via T3SS2, devoid of flagella and exposed to limiting zinc and magnesium concentrations. By contrast, cytosolic bacteria highly express genes implicated in iron uptake/storage, manganese and sugar transport suggesting that accumulation of these nutrients is important for bacterial proliferation in this compartment. Up-regulation of a subset of genes associated with oxidative stress resistance further indicates that bacteria are exposed to reactive oxygen species. Cytosolic bacteria are also T3SS1-active, flagellated and decorated with the MUC1-binding adhesin, SiiE, and therefore primed to adhere to, and enter, naïve cells upon their release.

and transport loci are up-regulated in cytosolic *S.* Typhimurium (Table 1 and Fig 2). *S. flexneri* uses a similar iron-scavenging strategy involving the synthesis of another siderophore, aerobactin [102], and the genes required for aerobactin synthesis and binding (*iucABCD* operon) are highly up-regulated in the cytosol (Table 1). The limited data available for UPEC IBCs are consistent with an important role of ferric iron acquisition for this bacterium as well (Table 1). Genes encoding the ferrous ($Fe^{2+}$) iron transport systems of *S. flexneri* (*feoB*, *sitABCD* and *mntH*) and *S.* Typhimurium (*sitABCD* and *mntH*) are also up-regulated, but not *feoB* (Table 1). The Sit and MntH systems in *S.* Typhimurium transport both ferrous iron and manganese, but likely function as manganese transporters under physiological conditions [59,60]. We found that a *S.* Typhimurium Δ*sitA*Δ*mntH* mutant is compromised for growth in the epithelial cytosol (Fig 7), indicating a specific requirement for $Mn^{2+}$ acquisition in this compartment, but the same transport systems do not contribute to the growth of *S. flexneri* in epithelial cells [92]. The plethora of iron acquisition strategies used by enteric bacteria means that genetic deletions in multiple pathways are required to affect intracellular proliferation. For

**Table 1. Cytosol signature genes.**

| Gene(s) | Function | Up-regulated in epithelial cytosol? | | |
|---|---|---|---|---|
| | | S. Typhimurium | S. flexneri[a,b,c,d] | UPEC[e,f] |
| *bioA* | Biotin synthesis | No | Yes | |
| *cysP* | Thiosulfate binding protein | Yes | No (downregulated) | |
| *entC* | Siderophore biosynthesis | Yes | Pseudogene | |
| *entF* | Enterobactin synthesis | Yes | Yes | Yes |
| *feoB* | Ferrous iron transporter | No | Yes | |
| *fepA* | Ferri-enterobactin receptor | Yes | Yes | Yes |
| *fepB* | Enterobactin binding protein | Yes | Pseudogene | |
| *fhuA* | Ferrichrome receptor | Yes | Yes | |
| *fhuE* | Ferric-coprogen transporter | Yes | Pseudogene | |
| *foxA* | Ferrioxamine B transporter | Yes | Gene is absent | |
| *fruBKA* | Fructose transport and phosphorylation | Yes | Yes | |
| *grxA* | Glutaredoxin 1 | Yes | Yes | |
| *iroBCDE* | Salmochelin synthesis, export and utilization | Yes | Genes are absent | |
| *iroN* | Siderophore receptor | Yes | Gene is absent | Yes |
| *iucABCD* | Aerobactin synthesis | Gene is absent | Yes | |
| *iutA* | Aerobactin receptor | Gene is absent | Yes | |
| *lysA* | Lysine biosynthesis | No | Yes | |
| *mgtA* | $Mg^{2+}$ transport ATPase | No | Yes | |
| *mntH* | $Mn^{2+}/Fe^{2+}$ transport | Yes | Yes | |
| *nrdHIEF* | Ribonucleotide reductases | Yes | Yes | |
| *phoRB* | Regulation of pho regulon | No | Yes | |
| *proVWX* | Glycine, betaine, proline transport | No | Yes | |
| *pstSC* | Phosphate acquisition | No | Yes | |
| *sfbABC* | ABC-type transporter | Yes | | |
| *sitABCD* | $Mn^{2+}/Fe^{2+}$ transport | Yes | Yes | Yes |
| *soxS* | Regulation of superoxide response regulon | Yes | Yes | |
| *sufA* | Fe-S cluster maturation protein | Yes | Yes | Yes |
| *uhpT* | Sugar phosphate transporter | Yes | Yes | |
| *yjjZ* | Unknown function | Yes | Yes | |

[a][90];

[b][91];

[c][92];

[d][93];

[e][94];

[f][95]

example, single gene deletions of *sitA*, *iucB* or *feoB* in *S. flexneri* have no effect on intracellular growth but a mutant in all three genes is unable to grow [103]. Similarly, *S.* Typhimurium Δ*entC* (defective in the synthesis of enterobactin and salmochelin; [104]) and Δ*fepB* (defective for the import of enterobactin, salmochelin and catecholate breakdown products containing iron; [96]) mutants have impaired cytosolic replication abilities (Fig 7).

We identified nineteen other signature genes of cytosolic colonization: *fruBKA*, *grxA*, *nrdHIEF*, *sfbABC*, *sufABCDSE*, *yjjZ* and *uhpT* (Table 1). GrxA, or glutaredoxin 1, is a redox-active protein [85] and NrdH is a glutaredoxin-like protein [105]. NrdEF is a $Mn^{2+}$-dependent ribonucleotide reductase that utilizes NrdH and glutaredoxin 1 (but not thioredoxin) as

hydrogen donors and functions under aerobic conditions, but only during iron restriction [106,107]. The *suf* operon encodes for an iron-sulfur cluster system and is induced by oxidative stress and iron starvation [108], like the *nrdHIEF* operon [109,110]. Notably, genes encoding for antioxidant proteins involved in bacterial resistance to hydrogen peroxide ($H_2O_2$), including *dps*, *katG*, *ahpC* and *ahpF* [111], were not induced in the epithelial cytosol. This finding suggests that superoxide anion generation from $H_2O_2$ by epithelial cells is not a major contributor to the oxidative stress-driven responses of *S.* Typhimurium in the cytosol. While the function of YjjZ remains unknown, *yjjZ* (SL1344_4483) is part of the Fur regulon in *S.* Typhimurium [112] and UPEC [113]; Fur is the master regulator of iron homeostasis in bacteria [114]. Transcription of *nrdHIEF*, *sufABCDSE* and *yjjZ* is therefore controlled by common factors, iron deprivation and Fur. Two genetic loci involved in sugar uptake or conversion, *fruBKA* and *uhpT*, were also identified as part of the cytosol transcriptional signature (Table 1 and Fig 8). *uhpT* and *fruBKA* are not induced by iron deprivation or oxidative stress but rather by sugars present in the eukaryotic cytosol, specifically glucose-6-phosphate [90] and fructose [84], respectively.

The combination of available transcriptome data revealed bacterial species-specific cytosol-induced genes. Specifically, *bioA*, *lysA*, *pstSC*, *phoRB*, *proVWX* and *mgtA* were induced in *Shigella* but not in *S.* Typhimurium (Table 1 and S1 Dataset; [91,103]). The differential expression profile suggests variance in the metabolic and phosphate acquisition strategies used by these two pathogens in the epithelial cytosol. The contribution of $Mg^{2+}$ transport also seems to differ between the two bacteria, as *mgtA*-encoded $Mg^{2+}$ acquisition is induced by the cytosolic environment for *S. flexneri* and not for *S.* Typhimurium. Unlike *S. flexneri*, *S.* Typhimurium has a second $Mg^{2+}$ transport system encoded by *mgtB*, which is up-regulated in the vacuole (Fig 5 and S1 Dataset). Furthermore, the third $Mg^{2+}/Ni^{2+}/Co^{2+}$ transporter (CorA) is constitutively expressed in *S.* Typhimurium [115,116] and is required for optimal proliferation in the cytosol [117]. The importance of $Mn^{2+}$ acquisition also seems to differ (see above).

We identified *S.* Typhimurium-specific coding and non-coding genes that serve as signatures of the cytosolic lifestyle. Examples include STnc3080, STnc3250, *SL1802* and *siiABCDEF*, as well as eight type III effectors that are translocated by T3SS1 (Figs 3, 6 and S3 and S1 Dataset). STnc3080, STnc3250 and *SL1802* are all negatively regulated by Fur [35,112,118] but their role in *Salmonella* pathogenesis remains unknown. Intracellular expression of the *siiABCDEF* operon has not been reported before, prompting us to establish that the SiiE protein, which mediates adhesion to the MUC1 on the apical surface of intestinal epithelial cells [119], is both on the surface of cytosolic bacteria and secreted into the cytosol of epithelial cells (S7 Fig). The combination of findings from this study, and others [12,36], shows that cytosolic *S.* Typhimurium are induced for T3SS1, secrete T3SS1-associated effectors, are flagellated and decorated with SiiE. We conclude that cytosolic *S.* Typhimurium are "primed" to enter naïve enterocytes following cellular release.

The major environmental factor that controls the transcriptional program of cytosol-residing bacteria is clearly iron-limitation. Within mammalian cells, the "labile iron pool" refers to the pool of chelatable and redox-active iron [120,121]. In resting cells, the labile iron pool is estimated to be ~1 μM, representing only a minor fraction of the total cellular iron (50–100 μM) [120,121]. At a subcellular level, measurements have shown that the concentration of labile iron is lower in the cytosol than the nucleus, mitochondria or endosomes/lysosomes [122,123], in agreement with our findings and those of others [67]. We believe that consumption of free iron from the cytosol by rapidly growing *S.* Typhimurium further depletes the pool compared to uninfected/resting cells.

The uniform induction of iron-responsive *S.* Typhimurium genes/sRNAs in the cytosol (Figs 3 and S3) indicates a generalized bacterial response to iron deprivation. Fur is the major

iron-responsive transcriptional regulator that controls iron homeostasis and oxidative stress defenses in bacteria [114]. When bound to iron (i.e. in iron-replete conditions), Fur acts as both a repressor and activator of gene expression, either directly or indirectly, and modulates expression of ~10% of the *S.* Typhimurium genome [112,124]. Except for *SL3990*, all the low $Fe^{2+}$/bile shock-regulated "up cytosol" genes/sRNAs (S1 Dataset) that were confirmed using transcriptional reporters (Figs 3 and S3) are negatively regulated by Fur [101,112,125] (Fig 2). Many have a Fur box in their promoter regions, the regulatory site to which iron-bound Fur binds, suggesting direct regulation. Of note, *sitABCD* and *mntH* belong to the MntR (SL1344_0810) mini-regulon that responds to manganese abundance, and are also regulated by Fur [53,56,126]. An unanswered question is why all Fur-repressed *S.* Typhimurium genes are not induced in the cytosolic population ([112]; S1 Dataset). It is also unclear why SPI1-associated genes are induced in a subset of cytosol-exposed bacteria in an iron-limited environment, when expression of the SPI1 regulon is typically activated by iron-bound Fur under iron-replete conditions via HilD and H-NS [127,128]. The regulation of SPI1 and its associated genes is complex [47], with numerous environmental signals feeding into the regulatory network including osmolarity, oxygen and bacterial growth phase, as well as iron. Bacterial cell-to-cell variation in the integration of all the relevant signals could explain the phenotypic heterogeneity observed for SPI1-associated gene expression in cytosolic bacteria (Fig 6).

We have identified some of the site-specific cues encountered by *S.* Typhimurium in the distinct milieus of the SCV versus the cytosol. *S.* Typhimurium residing within the epithelial cytosol have limited access to iron and manganese, are experiencing oxidative stress and utilize sugars as a carbon source. Interestingly, *S.* Typhimurium genes involved in iron uptake and storage, manganese uptake and Fe-S cluster synthesis are upregulated in the lag-phase of growth [129], which suggests a requirement for these particular genes as bacteria prepare for exponential growth in the mammalian cytosol [12]. Other studies of *S. flexneri* and *L. monocytogenes* colonization further describe the mammalian cytosol as a nutrient-rich environment of neutral pH with low concentrations of $Na^{2+}$, $Mg^{2+}$, $Ca^{2+}$ and $PO_4^{3-}$ ions, but a high concentration of $K^+$ [89,91,130,131]. In contrast, the mature SCV is acidic [7,132,133], has high levels of $K^+$ and oxygen, but is limiting in $Mg^{2+}$, $PO_4^{3-}$, and $Zn^{2+}$ and *S.* Typhimurium are experiencing oxidative stress but not amino acid starvation (Figs 3, 5 and 8) [42,43,67,91,134].

Based on single-cell analysis, we can refine previous conclusions drawn from population-based studies of *S.* Typhimurium and epithelial cells. For example, in the landmark study of Hautefort *et al.* (2008) [33], the simultaneous induction of SPI2, SPI1 and flagella within epithelial cells was reported, which at the time was puzzling. Single-cell microscopic analysis has solved this mystery by showing that the cytosolic population is solely responsible for the intracellular expression of both SPI1 and flagella ([12,14,16,36]; this study). Based on the intracellular induction of *iroA-* and *mgtB-lacZ* reporter fusions in MDCK (epithelial) cells, concentrations of $Fe^{2+}$ and $Mg^{2+}$ were thought to be low in the mature vacuole [135]. Additionally, a recent proteomic analysis of *S.* Typhimurium isolated from infected epithelial cells at 6 h p.i. suggested that bacteria are starved for numerous metals, including iron, manganese, and zinc [136]. Our findings show that metal limitation experienced by intracellular bacteria is niche-dependent; levels of iron and manganese are lower in the cytosol, whereas zinc and magnesium are more limiting in the vacuole (Fig 8), which demonstrates the importance of verifying results derived from population-based studies at the single-cell level. We suggest that the many studies that have identified individual *S.* Typhimurium or mammalian genes that influence bacterial proliferation in epithelial cells should be revisited, especially if the distinct replication niches were not considered. As heterogeneity in terms of bacterial proliferation and intracellular colonization site clearly influences the outcome of infection [137,138], it is crucial that the study of host-pathogen interactions incorporates single-cell analysis.

## Materials and methods

### Bacterial strains and plasmids

*S.* Typhimurium SL1344 was the wild-type strain used in this study [139]. Gene deletion mutants were constructed using *sacB* allelic exchange, λ red recombinase technology or P22 transduction. SL1344 Δ*sitA* (deleted for amino acids 4–298), Δ*mntH* (deleted for amino acids 1–411), Δ*yjjZ* (ΔSL1344_4483, deleted for amino acids 4–76) and ΔSL1344_1802 (deleted for amino acids 4–53) deletion mutants were made via *sacB* negative selection. Two fragments of ~1 kb upstream and downstream of the gene of interest were amplified from *S.* Typhimurium SL1344 genomic DNA using Phusion High-Fidelity DNA polymerase (Thermo Scientific) (primer sequences are listed in S1 Table). The two fragments were combined for a second round of amplification by overlap extension PCR. The amplicon was then digested with appropriate restriction enzymes, ligated into the suicide vector pRE112 (Cm^R) [140], and electroporated into *E. coli* SY327λpir. After sequence confirmation, the pRE112 plasmids were transferred to *E. coli* SM10λpir (Kan^R) for conjugation into SL1344 wild-type or Δ*mntH* bacteria (for the Δ*sitA*Δ*mntH* double mutant). For the second recombination event, *sacB*-based counterselection on LB agar containing 5% (w/v) sucrose was used, and streptomycin-resistant, chloramphenicol-sensitive colonies were screened by PCR with primers outside of the recombination region to confirm the deletion of each gene. *S.* Typhimurium SL1344 Δ*sufABCDSE*::kan, Δ*nrdHIEF*::kan, Δ*fhuB*::kan and Δ*fepB*::kan strains were generated using λ red recombineering technology [141]. λ red cassettes were amplified using pKD4 as a template with the oligonucleotide pairs listed in S1 Table and electroporated into *S.* Typhimurium SL1344 wild-type containing the temperature-sensitive helper plasmid, pKD46. Transformants were selected on LB agar plates containing streptomycin (100 μg/ml) and kanamycin (50 μg/ml). The correct integration of the kanamycin resistance cassette was confirmed by colony PCR with a target-flanked primer and a kanamycin resistance gene-specific primer. Each mutation was transferred into a clean SL1344 wild-type background using P22 phage transduction. The following insertional mutants were constructed by P22 transduction into a SL1344 wild-type background from phage lysates prepared from the following *S.* Typhimurium mutants: Δ*cirA*::kan, Δ*cysP*::kan, Δ*entC*::kan, Δ*fepA*::kan, Δ*fhuE*::kan, Δ*foxA*::kan, Δ*grxA*::Cm, Δ*iroBC*::kan, Δ*iroN*::kan, Δ*lpxR*::kan, Δ*mgtB*::kan, Δ*sfbA*::kan, Δ*siiABCD*::kan, ΔSL3990::kan, Δ*soxS*::kan, ΔSTnc3080::kan, Δ*uhpT*::kan, Δ*znuA*::kan ([96,142–144]; or provided by Manuela Raffatellu or Carsten Kroeger). Selection was on LB agar plates containing kanamycin (50 μg/ml) or chloramphenicol (30 μg/ml).

For genetic complementation, the deleted gene was either restored on a low copy number plasmid or in the chromosome. Specifically, Δ*entC* bacteria were transformed with pWSK29-*entCEBA* (provided by Steve Libby and Joyce Karlinsey), Δ*grxA*Δ*soxS* bacteria with plasmid-borne *soxS* under the control of its native promoter (pWKS30-*soxS*), Δ*mntH*Δ*sitA* bacteria with pWKS30-*sitABCD* [145] and Δ*znuA* bacteria with pWKS30-*znuA*, which contains a DNA fragment encompassing the *znuA* promoter region and coding sequence. The Δ*fepB* mutant was complemented with a chromosomal copy of *fepB* by transposition at the *att*Tn7 site [146]. Oligonucleotides used to construct pWKS30-*soxS*, pWKS30-*znuA* and pGP-Tn7-Cm-*fepB* are listed in S2 Table.

SL1344 wild-type bacteria constitutively expressing *mCherry* from the chromosome (*glmS*::*Ptrc-mCherryST*::FRT, denoted as mCherry-*S.* Typhimurium) or a plasmid (pFPV-mCherry) have been described previously [7,147]. *mCherry* expression is driven by the constitutive *trc* promoter or *S.* Typhimurium *rpsM* promoter, respectively. The *PuhpT-gfpova* plasmid, pNF101 [24], was used as a biosensor for bacterial exposure to the mammalian cytosol. Expression of the unstable GFP variant, *gfp_ova* (protein half-life <60 min, [148]), is under the

control of the glucose-6-phosphate responsive *uhpT* promoter from *S. flexneri*. Alternatively, bacterial access to the cytosol was assessed using the bidirectional reporter plasmid, pCHAR--Duo(ASV), whereby constitutive *mCherry* expression is driven by the synthetic ProB promoter and *gfpmut3.1*(ASV) expression is under the control of the *S.* Typhimurium *uhpT* promoter. To create pCHAR-Duo(ASV), the coding sequence for the stable GFP variant, *gfpmut3.1* (protein half-life >24 h), was excised from pCHAR-Duo [32] by XmaI/ApaI digestion and replaced with the coding sequence for a destabilized GFP variant, *gfpmut3.1*(ASV) (protein half-life ~110 min), which had been released from pGFP(ASV) (Clontech) by XmaI/ApaI digestion.

To construct fluorescent transcriptional reporters, we extracted the precise transcriptional start sites from the online SalComMac database (http://tinyurl.com/SalComMac; [34,43]). Approximately 500 bp of sequence upstream of the ATG start codon of each gene, or transcript start site of each sRNA, was amplified from SL1344 genomic DNA with the oligonucleotide pairs listed in S3 Table. Amplicons were digested with XbaI/SmaI and ligated into the corresponding restriction sites of pGFPmut3.1 (Clontech). XbaI/ApaI digestion released the gene/sRNA promoter-*gfpmut3* fragment, which was then ligated into the corresponding sites of the low copy number plasmid, pMPMA3ΔPlac [70]. The *pipB* and *sopB* promoters were excised from pMPMA3ΔPlac-P*pipB*-GFP(LVA) and pMPMA3ΔPlac-P*sopB*-GFP(LVA), respectively [70], by digestion with XbaI/KpnI and ligated into the corresponding sites of pGFPmut3.1 and then transferred to pMPMA3ΔPlac as described above. The pMPMA3ΔPlac-P*uhpT*-*gfpmut3* construct has been described previously [149]. GFP reporter plasmids were electroporated into wild-type mCherry-*S.* Typhimurium.

## Reporter assays for metal sensitivity

For metal repression studies, wild-type mCherry-*S.* Typhimurium bacteria harboring *gfpmut3* fusions were grown shaking overnight (220 rpm) at 37˚C in LB-Miller broth containing 100 μg/ml streptomycin and 50 μg/ml carbenicillin, then washed twice in M9 salts pH 7.0 (6 g $Na_2HPO_4$, 3 g $KH_2PO_4$, 0.5 g NaCl, 1 g $NH_4Cl$ per liter) supplemented with 1 mM $MgSO_4$, 0.4% (w/v) glucose and 0.01% (w/v) histidine–hereafter referred to as M9-supplemented media–and resuspended in an equal volume of M9-supplemented media. Washed bacteria were diluted 1:200 in M9-supplemented media containing 100 μg/ml streptomycin and 50 μg/ml carbenicillin and the desired concentration of cobalt (II) chloride (Alfa Aesar, Puratronic, 99.998%), iron (III) chloride (Acros Organics, 99+%), manganese (II) chloride (Alfa Aesar, Puratronic, 99.999%), nickel (II) chloride (Alfa Aesar, Puratronic, 99.9995%) or zinc chloride (Alfa Aesar, Puratronic, 99.999%). All solutions were prepared fresh in MilliQ water. Bacterial cultures (2 ml) were grown in 14 ml polystyrene round-bottom tubes (Falcon) at 37˚C, shaking at 220 rpm for ~16 h. GFP fluorescence was measured in black 96-well plates (Costar) using a TECAN SPARK plate reader (excitation wavelength of 485 nm, bandwidth 20 nm; emission wavelength of 535 nm, bandwidth of 20 nm). The background fluorescence of mCherry-*S.* Typhimurium bacteria (no reporter plasmid) was subtracted from all readings, which were then normalized for bacterial growth ($OD_{600}$).

## Measurement of bacterial growth in broth

Bacterial cultures were grown overnight in LB-Miller broth, shaking (220 rpm) at 37˚C, then subcultured 1:100 into 10 ml LB-Miller broth in 125 ml Erlenmeyer flasks with shaking (220 rpm) at 37˚C. Samples were collected at hourly intervals for $OD_{600}$ readings.

## Mammalian cell culture

HeLa human cervical adenocarcinoma epithelial cells were purchased from American Type Culture Collection (ATCC, CCL-2) and used within 15 passages of receipt. Cells were

maintained at 37°C and 5% $CO_2$ in the growth medium (GM) recommended by ATCC i.e. Eagle's minimum essential medium (EMEM, Corning) containing sodium pyruvate, L-gluta-mine and 10% (v/v) heat-inactivated fetal calf serum (FCS; Gemini Bio Products). Tissue culture plasticware was purchased from Thermo Scientific Nunc.

### Bacterial infection of mammalian cells

HeLa cells were seeded 24 h prior to infection at the following densities: (i) $5x10^4$ cells/well in 24-well tissue culture plates, (ii) $6x10^4$ cells/well on acid-washed glass coverslips in 24-well plates, or (ii) $2.6x10^6$ cells/15 cm tissue culture dish. T3SS1-induced bacterial subcultures were prepared in LB-Miller broth (Difco) [150] and cells were infected for 10 min with bacterial subcultures (MOI ~50) as described [150]. To induce autophagy, HeLa cells were shifted to Earle's Balanced Salt Solution (EBSS, Sigma) 3 h prior to infection and maintained in EBSS until 90 min p.i. Thereafter, infected cells were switched back to regular growth medium. To inhibit autophagy, cells were incubated in growth media containing 100 nM wortmannin (WTM, Calbiochem) for 45 min prior to infection, continuing to 90 min post-infection, whereupon cells were transferred back to regular growth media. For manipulation of cellular cation levels, epithelial cells were treated 16–18 h prior to infection in growth media containing 200 μM 2,2'-dipyridyl (DPI, Sigma-Aldrich), 200 μM DPI and 200 μM ammonium iron (III) citrate (FAC, Alfa Aesar) or 200 μM FAC alone. DPI +/- FAC treatment continued throughout the infection. DPI was prepared as a 100 mM stock in 95% ethanol and stored at -80°C for a maximum of 2 weeks. FAC was prepared extemporaneously in MilliQ water. Gentamicin protection and CHQ resistance assays (chloroquine diphosphate salt, Sigma) were as described previously [150]. Monolayers were solubilized with 0.2% (w/v) sodium deoxycholate (Sigma) and serial dilutions were plated on LB-Miller agar (Remel) for enumeration of colony forming units (CFUs).

### Bacterial RNA enrichment from infected cells, cDNA synthesis and Illumina sequencing

*S.* Typhimurium was enriched from infected HeLa cells (12 x 15 cm dishes for EBSS treatment, 6 x 15 cm dishes for WTM treatment) as described previously by Hautefort *et al.* (2008) [33] with minor modifications. Briefly, infected HeLa cells were washed once with 30 ml chilled phosphate-buffered saline (PBS), then lysed on ice for 10–20 min in 0.1% SDS, 1% acidic phenol, 19% ethanol in water (15 ml per 15 cm dish). Pooled lysates were collected into 50 ml conical tubes and centrifuged at 3,220 xg for 30 min, 4°C. After three washes in ice-cold wash buffer (0.1% phenol, 19% ethanol in water) (30 min, 3,220 xg, 4°C each wash), the remaining bacterial pellet was resuspended in <1 ml of wash buffer, transferred to an RNase-free micro-centrifuge tube and centrifuged at 16,000 xg for 2 min. The supernatant was discarded, and the pellet resuspended in 1 ml TRIzol reagent (Life Technologies) by gently pipetting up-and-down (~60 times), then transferred to -80°C for storage. Total RNA was extracted as described [34] and the RNA concentration quantified using a Nanodrop 2000 spectrophotometer (Thermo Scientific). RNA quality was analyzed using an Agilent Bioanalyzer 2100.

RNA samples isolated from two independent experiments were sent to Vertis Biotechnologie AG (https://www.vertis-biotech.com) for cDNA synthesis and RNA sequencing. The RNA samples were fragmented with ultrasound (4 pulses of 30 sec at 4°C) followed by a treatment with antarctic phosphatase and re-phosphorylated with polynucleotide kinase. Afterwards, the RNA fragments were poly(A)-tailed using poly(A) polymerase and an RNA adapter was ligated to the 5'-phosphate of the RNA. First-strand cDNA synthesis was performed using an oligo(dT)-adapter primer and M-MLV reverse transcriptase. The resulting cDNA was PCR-

amplified to about 20–30 ng/μl using a high-fidelity DNA polymerase. The cDNA was purified using the Agencourt AMPure XP kit (Beckman Coulter Genomics) and analyzed by capillary electrophoresis. The primers used for PCR amplification were designed for TruSeq sequencing according to the instructions of Illumina. The four cDNA samples were pooled in equimolar amounts and size fractionated in the size range of 150–500 bp using a differential clean-up with the Agencourt AMPure kit, then the cDNA samples were sequenced on an Illumina HiSeq 2000 machine with 50 bp read length.

## RNA-seq data analysis

A total of 42 to 57 million sequence reads were generated from each sample. Alignment and processing of bacterial RNA-seq reads was done using bacpipe pipeline v0.8a (https://github.com/apredeus/multi-bacpipe). Briefly, reads were aligned to a reference genome sequence of S. Typhimurium strain 4/74 that included the complete chromosome, as well as 93 kb pSLT virulence plasmid, 87 kb pCol1B9 plasmid, and 9 kb pRSF1010 plasmid (NCBI assembly GCF_000188735.1, replicon GenBank IDs CP002487.1—CP002490.1). Reads were aligned using STAR v2.7.6a with "—alignIntronMin 20—alignIntronMax 19—outFilterMultimapN-max 20" options. Ribosomal operons were predicted using Prokka v1.14.6, after which reads overlapping rRNA operons were removed from the bam file using bedtools v2.29.2 with "bedtools intersect -nonamecheck -v" options. Remaining reads were subsequently quantified using the featureCounts program from Subread v2.0.1, with "-O -M—fraction -t gene -g ID -s 1" options. Custom GTF annotation of the 4/74 strain with 282 known *Salmonella* non-coding RNAs was used and is available at https://github.com/apredeus/salmonella_pathways/tree/main/474_ref. From 51 to 59% of reads aligned to rRNA operons; of the remaining reads, 11.7–21.6 M aligned to the reference genome, and 10.1–18.5 M (22.7–33.9%) were assigned to an annotated gene. Raw sequencing reads have been deposited at Gene Expression Omnibus (GEO) database under series ID GSE179103. Differential expression analysis was done in R v4.0.4 using DESeq2 v1.30.1. Genes with adjusted p-value of 0.05 or below were deemed differentially expressed. TPM values were calculated in bacpipe v0.8a (see above). Plotting was done using ggplot2 v3.3.3.

## Pathway enrichment analysis

A collection of 80 custom, *Salmonella*-specific pathways was generated from previous RNA-seq, ChIP-seq, and microarray studies of S. Typhimurium regulons [112,124,128,151,152], as well as transcriptional profiling of S. Typhimurium 4/74 under infection-relevant *in vitro* growth conditions [34] and inside murine macrophages [43]. A detailed description of the pathway curation process and all relevant scripts can be found at https://github.com/apredeus/salmonella_pathways. Briefly, custom curated gene sets of SPI1 plus SPI1-translocated effectors, SPI2 plus SPI2-translocated effectors, and iron transporters and siderophores were compiled using many literature sources that are listed in the "Salmonella custom pathways.xlsx" table in the repository above (supp_tables subdirectory). For data from [112,151,152] [124,128], supplementary tables from the publications with differentially expressed genes were used; up- and down-regulated genes were used to create separate pathways. Whenever the authors used a different reference strain, gene IDs were converted to 4/74 IDs using the provided ortholog table. For single-replicate RNA-seq experiments profiling 4/74 and D23580 under the seventeen *in vitro* conditions, the following strategy was used. Tables of TPM-normalized expression values (see supp_tables/474.tsv, D23.tsv) were used to calculate fold change (FC) of a particular condition with regard to all 16 others. Genes with log2FC of 2 or more were selected as "marker" genes for a particular *in vitro* condition; only up-regulated genes

were considered for this specific dataset. The lists of genes up-regulated in each single-replicate experiment were generated using the calculate_fc_tpm.pl script available from the same repository. Additionally, KEGG pathways for *S.* Typhimurium SL1344 were obtained using the KEGGREST R package v1.28.0, and gene ontology (GO) mapping of SL1344 genes was extracted from SL1344 protein annotation by InterProScan v5.41–78. See S2 Dataset for the entire dataset. The combined pathways were used to annotate gene sets up-regulated in the vacuole and cytosol using the clusterProfiler R package v3.16.1. The resulting overlaps were visualized using ggplot2 v3.3.2.

## Fluorescence microscopy

HeLa cells were seeded on acid-washed, 12 mm glass coverslips (#1.5 thickness, Fisher Scientific) in 24-well plates. Infected HeLa cells were fixed with 2.5% (w/v) paraformaldehyde in PBS for 10 min at 37˚C. The immunostaining procedure has been described previously [24]. Rabbit anti-SiiE serum (kindly provided by Michael Hensel) was used at a dilution of 1:200. Digitonin permeabilization to deliver anti-*Salmonella* LPS antibodies (*Salmonella* O-Antiserum Group B Factors 1, 4, 5, 12; Difco; 1:300 dilution) directly to the cytosol was as described previously [14]. Cells were stained with Hoechst 33342 (1:10,000 in DDH$_2$O, Life Technologies) to label DNA and coverslips were mounted onto glass slides using Mowiol.

Samples were visualized on a Leica DM4000 upright fluorescence microscope for scoring the number of bacteria per cell, the proportion of cytosolic bacteria and quantification of transcriptional reporter activity. ImageJ software was used to quantify the activity of transcriptional reporters at the individual bacterium level. Images in the red (mCherry) and green (GFP) channels were acquired sequentially under 63x magnification on a Leica DM4000 upright fluorescent microscope using the same acquisition settings (exposure and gain) for each group of transcriptional reporters (determined by the reporter with the highest intensity in the green fluorescence channel). A minimum of 100 cytosolic and 100 vacuolar bacteria, from cells with ≥100 bacteria and 2–40 bacteria, respectively, acquired from ≥5 random fields of view were quantified for each transcriptional reporter from each experiment. From grayscale images, well-defined, individual bacteria were arbitrarily chosen on the mCherry channel (represents all bacteria) and manually outlined, converted to a binary image and the pixel intensity associated with each identified particle (i.e. one bacterium) quantified from the corresponding GFP channel image. GFP fluorescence/bacterium was subsequently plotted as the mean fluorescence intensity (MFI).

Images were acquired using confocal microscopy (Leica SP8) in sequential acquisition mode through an optical section of 0.3 μm in the z-axis. Images are maximum intensity projections of z-stacks.

## Statistical analysis

All experiments were conducted on at least three separate occasions, unless otherwise indicated, and results are presented as mean ± SD. Except for the RNA-seq data (see separate paragraph on RNA-seq data analysis), statistical analyses were performed using: (i) one-way analysis of variance (ANOVA) with Dunnett's post-hoc test, or (ii) one-way ANOVA on ranks (Kruskal-Wallis test) for data with a non-normal distribution (i.e. number of bacteria/cell, MFI) (GraphPad Prism). A p-value of <0.05 was considered significant.

## Supporting information

**S1 Fig. Enrichment of bacterial RNA from infected cells.** Total RNA was extracted from HeLa epithelial cells (HeLa), wild-type *S.* Typhimurium SL1344 grown to late log-phase in

LB-Miller broth (STm), *S.* Typhimurium isolated from Earle's balanced salt solution (EBSS)-treated cells at 8 h p.i. or *S.* Typhimurium isolated from wortmannin (WTM)-treated cells at 8 h p.i. RNA quality was analyzed by electrophoretic separation using an Agilent Bioanalyzer 2100. Ladder sizes shown in kb.
(TIF)

**S2 Fig. Relative expression of the different PAIs of *S. Typhimurium*.** Each arrow represents an individual gene to scale within each PAI. The different islands are also scaled against each other. The color of each arrow represents relative gene expression–red arrows depict genes up-regulated in the cytosol (≥1.40-fold change WTM/EBSS), blue are genes up-regulated in the vacuole (≥1.40-fold change EBSS/WTM), yellow are genes with unchanged expression (0.72–1.39-fold change) and grey arrows are genes with a TPM value <10 and considered not expressed. See S1 Dataset for the entire data set. Adapted from Srikumar et al. [43].
(PPTX)

**S3 Fig. Additional cytosol-induced iron-associated genes.** Epithelial cells were infected with mCherry-*S.* Typhimurium harboring *gfpmut3* transcriptional reporters. At 8 h p.i., cells were fixed & stained with Hoechst 33342 to detect DNA. Representative confocal microscopy images show induction of *mntH*, *entC*, *SL3990*, STnc3250, *fhuE*, *SL1802*, *iroN*, *fepB*, *fepA* and *fhuA* promoters in cytosolic bacteria. Green = transcriptional reporter, red = *S.* Typhimurium, blue = DNA. Scale bars = 10 μm.
(TIF)

**S4 Fig. RNA-seq-predicted genes that were not confirmed by transcriptional reporters.** Epithelial cells seeded on coverslips were infected with mCherry-*S.* Typhimurium harboring *gfpmut3* transcriptional reporters. At 8 h p.i., cells were fixed and stained with Hoechst 33342 to detect DNA. Representative confocal microscopy images show equivalent expression of *asnA*, *ilvC*, *mtr*, *proV*, *SL1344_2715*, *ygbA*, STnc4000 and *trpE* promoters in vacuolar and cytosolic bacteria. Green = transcriptional reporter, red = *S.* Typhimurium, blue = DNA. Scale bars are 10 μm.
(TIF)

**S5 Fig. Metal responsiveness of transcriptional reporters.** (A). GFP fluorescence in mCherry-*S.* Typhimurium harboring P*iroN-gfpmut3*, P*sitA-gfpmut3*, P*yjjZ-gfpmut3*, P*sufA-gfpmut3*, P*fepA-gfpmut3* or P*STnc3250-gfpmut3* reporter plasmids. Bacteria were grown shaking overnight at 37°C for 16 h in M9-supplemented media containing increasing concentrations of $CoCl_2$, $FeCl_3$, $MnCl_2$, $NiCl_2$ or $ZnCl_2$ (0.1 μM, 1 μM, 10 μM or 100 μM). No added cation (0) served as the control. The relative fluorescence units (RFU) were normalized to $OD_{600}$ and expressed as a percentage of control (set to 100%). Background GFP fluorescence of mCherry-*S.* Typhimurium (no reporter) was subtracted from all values. n≥3 independent experiments. (B) GFP fluorescence in mCherry-*S.* Typhimurium harboring P*zinT-gfpmut3* or P*mgtC-gfpmut3* plasmids. Growth and analysis of *zinT* expression was as described in (A). Standard M9 minimal media contains 1 mM $MgSO_4$ which completely represses *mgtC* expression. The effect of $Mg^{2+}$ concentration on *mgtC* expression was therefore assessed in M9-supplemented media containing decreasing amounts of $MgSO_4$ i.e. 1000 μM (standard), 100 μM and 10 μM. $Mg^{2+}$ concentrations lower than 10 μM impacted bacterial growth. n = 3 independent experiments.
(TIF)

**S6 Fig. Growth curves of *S. Typhimurium* in rich media.** (A) Overnight cultures of wild-type *S.* Typhimurium were subcultured 1:100 in LB-Miller broth in the presence of 200 μM 2,2'-

dipyridyl (DPI) or vehicle (ethanol) control. Growth was measured every hour by optical density at 600 nm ($OD_{600}$). n = 3 independent experiments. (B) Overnight cultures of wild-type *S*. Typhimurium or the indicated deletion mutants were subcultured 1:100 in LB-Miller broth. Growth was measured every hour by optical density at 600 nm ($OD_{600}$). n = 3 independent experiments.
(TIF)

**S7 Fig. Cytosolic bacteria produce SiiE.** Epithelial cells seeded on coverslips were infected with mCherry-*S*. Typhimurium harboring a P*siiA-gfpmut3* transcriptional reporter. At 8 h p.i., cells were fixed and immunostained with polyclonal antibodies directed against SiiE. DNA was stained with Hoechst 33342. Representative confocal microscopy images show SiiE attached to (upper panel) or secreted by (lower panel) cytosolic bacteria. Inset shows enlargement of boxed area. Green = P*siiA-gfpmut3* reporter, red = *S*. Typhimurium (STm), white = SiiE, blue = DNA. Scale bars are 10 μm.
(TIF)

**S8 Fig. Identification of additional cytosol-induced *S. Typhimurium* genes.** Upper panels: Epithelial cells were infected with mCherry-*S*. Typhimurium harboring *gfpmut3* transcriptional reporters. At 8 h p.i., cells were fixed & stained with Hoechst 33342 to detect DNA. Representative confocal microscopy images show induction of *cysP*, *grxA*, *soxS*, *sfbA*, *fruB* and *uhpT* promoters in cytosolic bacteria. Green = transcriptional reporter, red = *S*. Typhimurium, blue = DNA. Scale bars are 10 μm. Lower panel: Quantification of the MFI of GFP signal by fluorescence microscopy and ImageJ. Small dots represent individual bacteria; large dots indicate the mean of each experiment; horizontal bars indicate the average of 2–3 experiments. Acquisition parameters (exposure time and gain) were set-up using P*cysP-gfpmut3* (the highest GFP signal intensity) and these same parameters were applied throughout. Dashed lines indicate the range of background fluorescence in the GFP channel measured for mCherry-*S*. Typhimurium (no reporter).
(TIF)

**S1 Table. Oligonucleotides used to construct gene deletion mutants.**
(DOCX)

**S2 Table. Oligonucleotides used for genetic complementation.**
(DOCX)

**S3 Table. Oligonucleotides used to construct *gfpmut3* transcriptional reporters.**
(DOCX)

**S1 Dataset. RNA-seq data set.** Tab 1 shows the raw counts. Raw sequencing reads have been deposited at Gene Expression Omnibus (GEO) database under series ID GSE179103. A total of 42 to 57 million sequence reads were generated from each sample. Tab 2 shows transcripts per million (TPM). All genes with mapped reads were reported as TPM and a threshold TPM value of 10 was used as a cut-off to distinguish gene expression from background. Tab 3 shows differential sequence analysis via DESeq2 v1.30.1. Tab 4 shows genes that were deemed differentially expressed (adjusted p-value of 0.05 or below). Tab 5 and Tab 6 shows the expression profiles of "up cytosol" and "up vacuole" genes that passed statistical significance in 22 distinct infection-relevant *in vitro* growth conditions plus in macrophages (http://bioinf.gen.tcd.ie/cgi-bin/salcom.pl?db=salcom_mac_HL).
(XLSX)

**S2 Dataset. The entire dataset used for the custom pathway enrichment analysis of "up cytosol" and "up vacuole" genes/sRNAs (relates to Fig 2B).**
(XLSX)

## Acknowledgments

We kindly thank Michael McClelland, Corrie Detweiler, Carsten Kroeger, Denise Monack, Martina Sassone-Corsi and Manuela Raffatellu for sharing bacterial mutants; Michael Hensel for providing the anti-SiiE antibody; Manuela Raffatellu, Steve Libby, Joyce Karlinsey, Dirk Bumann and Olivia Steele-Mortimer for providing plasmids; Isabelle Hautefort for invaluable advice about bacterial RNA purification; Andrea Battistoni for helpful discussions about zinc transporters and Jean Celli and Johanna Elfenbein for critical reading of this manuscript. We greatly appreciate the artistic skills of Eliza Wolfson. Lastly, we acknowledge Henrietta Lacks and her family for their generous contributions to the biomedical community.

## Author Contributions

**Conceptualization:** TuShun R. Powers, Leigh A. Knodler.

**Data curation:** Alexander V. Predeus, Karsten Hokamp.

**Formal analysis:** Alexander V. Predeus, Karsten Hokamp, Leigh A. Knodler.

**Funding acquisition:** TuShun R. Powers, Jay C. D. Hinton, Leigh A. Knodler.

**Investigation:** TuShun R. Powers, Amanda L. Haeberle, Zeus Saldaña-Ahuactzi, Leigh A. Knodler.

**Methodology:** TuShun R. Powers, Amanda L. Haeberle, Alexander V. Predeus, Disa L. Hammarlöf, Jennifer A. Cundiff, Zeus Saldaña-Ahuactzi, Karsten Hokamp, Leigh A. Knodler.

**Supervision:** Jay C. D. Hinton, Leigh A. Knodler.

**Visualization:** Alexander V. Predeus, Leigh A. Knodler.

**Writing – original draft:** Jay C. D. Hinton, Leigh A. Knodler.

**Writing – review & editing:** TuShun R. Powers, Amanda L. Haeberle, Alexander V. Predeus, Disa L. Hammarlöf, Jennifer A. Cundiff, Zeus Saldaña-Ahuactzi, Karsten Hokamp.

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
