## [Decision Letter · Decision Letter 0]

17 Feb 2021

Dear Prof Knodler,

Thank you very much for submitting your manuscript "Niche-specific profiling reveals transcriptional adaptations required for the cytosolic lifestyle of Salmonella enterica" for consideration at PLOS Pathogens. As with all papers reviewed by the journal, your manuscript was reviewed by members of the editorial board and by several independent reviewers. In light of the reviews (below this email), we would like to invite the resubmission of a significantly-revised version that takes into account the reviewers' comments.

You will see that all reviewers praise the relevance and quality of your work here and all deem the work revealing the different profiles of two important intracellular populations of Salmonella to be very interesting. They however note a few points that need attention that we encourage you to address.

We cannot make any decision about publication until we have seen the revised manuscript and your response to the reviewers' comments. Your revised manuscript is also likely to be sent to reviewers for further evaluation.

Sincerely,

Sophie Helaine

Associate Editor

PLOS Pathogens

Raphael Valdivia

Section Editor

PLOS Pathogens

Kasturi Haldar

Editor-in-Chief

PLOS Pathogens

orcid.org/0000-0001-5065-158X

Michael Malim

Editor-in-Chief

PLOS Pathogens

orcid.org/0000-0002-7699-2064

Reviewer's Responses to Questions

**Part I - Summary**

Reviewer #1: This study reports the niche-specific expression profile of the facultative intracellular pathogen Salmonella Typhimurium. Specifically, upon entry of host epithelial cells, S. Typhimurium is known to inhabit a vacuolar compartment (the SCV), but in some cell types, a small fraction of bacteria escape into the cytosol where they replicate to high numbers; a phenomenon counteracted by host xenophagy (a special form of autophagy). Knodler and colleagues combine experimental induction or inhibition of host autophagy to decrease or increase the cytosolic Salmonella fraction, respectively, with bulk RNA-seq of the recovered intracellular bacteria from the thus treated cell cultures. By comparing genome-wide expression between SCV-enriched and cytosol-enriched bacteria, they define gene sets selectively expressed by intracellular Salmonella dependent on the sub-cellular compartment they occupy. For example, vacuolar Salmonella induce genes that reflect zinc and magnesium limitation. Vice versa, cytosolic pathogens seem to be deprived of iron, upregulated invasion genes required for a second round of infection, and had an expression profile indicative of high proliferation. Using transcriptional reporter fusions in combination with fluorescence microscopy, niche-specific induction of a subset of these genes could be confirmed. What is more, metal ion supplementation assays confirmed the sensitivity of specific gene promoters to the absence of the predicted metal ions. Finally, defined Salmonella deletion mutants were used for infection assays to test for changes in the distribution of vacuolar vs. cytosolic bacteria.

Overall, the definition of the niche-specific transcriptome of this important model pathogen is interesting, relevant and timely. However, I have some technical concerns about the experimental design and analysis, and a couple of comments that I hope will improve this manuscript.

Reviewer #2: In this manuscript, Powers and collaborators successfully established, using RNA-sequencing subsequently validated by transcriptional fusions, the transcriptomic signatures of two Salmonella populations colonizing two different niches within epithelial cells: the cytosol and the vacuole. This impressive resource is very well written, clearly structured, and the hypothesis and results are backed up and put into perspective in a very complete discussion of the literature. To start with, the RNA-seq comparison between cytosolic and vacuolar enriched populations very clearly revealed two distinct virulence genetic programs of the bacteria present in the two different niches highlighted notably by differential metal deprivations (Fe and Mn for the cytosol and Zinc and Mg for the vacuole). The authors then validate the differential RNA-seq signatures using transcriptional fusions of the genes induced in bacteria present in those two niches. Altogether this transcriptional atlas of cytosolic and vacuolar Salmonella reveals what specific program is needed for bacteria to colonize one or the other niche and advances the comprehension of how bacteria can successfully colonize different niches within the host. Therefore, I recommend this paper for publication in Plos Pathogens as the quality of the manuscript and the data presented together with the host-pathogen interaction topic of the manuscript meet the standards of the journal. However, before publication, the authors should answer/take notice of minor concerns listed in the next session.

Reviewer #3: Please use this section to discuss strengths/weaknesses of study, novelty/significance, general execution and scholarship

In this manuscript by Powers et al, the authors start by manipulating the autophagy levels in epithelial cells to shift the intracellular Salmonellapopulation to obtain more (or less, depending on the treatment) cells containing cytosolic bacteria, to further study the transcriptional changes of these bacterial sub-populations by RNA-Seq. The authors then combine their RNA-Seq data with other expression data previously published in the literature, to define genes differentially up regulated either in vacuolar or cytosolic Salmonella. After that, the authors first define and confirm the importance of specific genes related to the differential presence of metals between the cytosol and the vacuole. Then, the authors analyze the expression of cytosol-specific genes that are also up-regulated in late exponential/early stationary phase in vitro. Among these, the authors highlight genes encoding structural genes of the T3SS-1 and some effectors translocated by this system, but also the adhesin SiiE and the lipid-A deacylase LpxR. Having identified the genes that are up-regulated in the cytosol, the authors then go on and use CHQ resistance assays to confirm that the deletion mutants of 5 (of the 18 candidates analyzed) show a reduced proportion of cytosolic bacteria, compared to the wild-type strain. Finally, the authors confirm the results on these 5 candidates using single-cell fluorescence microscopy.

This study is very interesting, as it demonstrates important pathways that define the transcriptional adaptation of Salmonellato these two very different intracellular niches. The experiments are very well-thought and technically sound, and the analyzed phenotypes are very consistent. The main strength is the study of expression of specific genes using single-cell analysis in the context of alterations of the available metals for intracellular bacteria.

The main weakness of the study is the lack of a dynamic study on how the studied pathways affect the intracellular sub-populations, since the authors in general focus on 7-8 hpi, when these sub-populations are already established, but it is not clear if the expression of any of these genes is what drives the establishment of the population or only a consequence. This is particularly interesting in the context of SPI-1 co-regulated genes, since it would be nice to see if the T3SS-1 and effectors are continuously “on” after invasion or if they are truly up-regulated once the bacterium reaches the cytosol.

**Part II – Major Issues: Key Experiments Required for Acceptance**

Reviewer #1: • The authors treat the epithelial cell cultures with either EBSS or WTM to induce or inhibit autophagy. Importantly, in both cases treatment of cells extends for 90 min after the bacteria were added to the cultures. Can the authors exclude that EBSS and WTM treatment itself have an effect on Salmonella gene expression? The cleanest way of controlling for this would be to treat an in vitro Salmonella culture with EBSS or WTM, respectively, and sequence their transcriptome. Given that the transcriptomic dataset is such a central part of this study and will likely serve as a resource for the field, I deem this an essential control experiment.

• The way the RNA-seq data are analyzed is – at least – uncommon in the field. The authors employ two different tools: RankProduct was originally developed for microarray data and, as far as I know, has previously not been employed to RNA-seq data. BitSeq is a tool to call differentially expressed transcript isoforms in eukaryotes. I do not understand the rationale behind using these two tools, in combination, for the present dataset, rather than using well-established, benchmarked tools that do exist and are commonly used by the community (like edgeR or DESeq). Also, the cutoffs applied seem arbitrary (e.g. fold-change ≤0.71) and are relatively loose (e.g. FDR of 0.2 means that the risk of false discovery is 20%, whereas commonly this cutoff is set to 0.05 for RNA-seq studies). Can the authors please elaborate?

• In their final experiment, the authors test the ability of defined Salmonella mutants to escape/proliferate in the mammalian cytosol (Fig. 7). In case of four mutants, they claim that the fraction of cytosolic bacteria is significantly lower than in wildtype infections. However, the effects are marginal (whether or not they are indeed statistically significant I cannot assess). Why don’t the authors test the performance of multiple (triple, quadruple) deletion mutants in this assay; this may result in stronger differences (in fact, the authors even mention this possibility in the discussion; lines472-474). Additionally, judged from Fig. 7B it seems as if the slightly reduced bacterial numbers in the cytosol (in Fig. 7A) are mostly due to a reduced rate of vacuolar escape (rather than lower proliferation in the cytosol). Have the authors tested, whether the respective mutants may simply show a delayed vacuolar escape as compared to wildtype bacteria? E.g. how would the result look like when samples are harvested at a later timepoint of infection? Additionally, to test for an intracellular-specific effect, standard growth curves of these mutant strains in rich medium should be reported (as a supplementary figure).

Reviewer #2: (No Response)

Reviewer #3: Please use this section to detail the key new experiments or modifications of existing experiments that should be absolutely required to validate study conclusions.

Generally, there should be no more than 3 such required experiments or major modifications for a “Major revision” recommendation. If more than 3 experiments are necessary to validate the study conclusions, then you are encouraged to recommend “Reject”

1) The results in this manuscript would benefit from some live microscopy (or at least previous time points) for the expression of genes cytosol- or vacuole-specific. A more dynamic view on the regulation of these pathways would nicely complement the results already shown in this manuscript. This would particularly be interesting for the “SPI-1 associated” genes, as described above. These experiments could easily be done using the gfpmut3constructs.

2) In the last section of the results, the authors use the population-based CHQ resistance assay and mutants in identified genes that are expressed in the cytosol, to evaluate their role in the colonization of this specific niche. The results show that deletion mutants of 5 of the 18 candidates show a reduced proportion of cytosolic bacteria. These results are complemented using the dual-fluorescence reporter pCHAR-Duo(ASV), but only for these 5 candidates. It would be great if the authors can repeat these experiments with one to threeof the 13 candidates that did not showed a change in the CHQ resistance assay compared to the wild-type strain, as even the authors comment that the single-cell analysis has a way better resolution than the population-based analysis. On the same line, it would be good to have some complementation assays for at least some the analyzed mutants that show phenotype.

3) Finally, regarding the pathway enrichment analysis, it would be great if the authors can provide more technical detail on how the analysis was performed, as the authors only link to a webpage containing the scripts. It would be nice to have a more detailed description in the manuscript on how the analysis was made.

**Part III – Minor Issues: Editorial and Data Presentation Modifications**

Reviewer #1: • The authors use the term “genes/sRNAs” throughout the manuscript. The reason for doing this remains unclear. The term “gene” includes both coding and noncoding genes. I suggest to replace “genes/sRNAs” with “genes” whenever they speak of expression, and with “mRNAs/sRNAs” whenever they refer to transcripts.

• At the beginning of the section describing the RNA-seq screen (line 166 ff.), the authors may want to give explicit numbers for the fraction of cytosolic/vacuolar bacteria in the EBSS- or WTM-treated samples that were subjected to RNA-seq. This may be somewhat inferred from Fig. 1C. However, given that the RNA-seq screen is such an integral part of this study, I think this should become clearer as it would otherwise mislead readers to think that the respective samples contained pure populations.

• For different assays, the authors use seemingly redundant reporter plasmids for cytosol-exposed bacteria, namely pNF101 and pCHAR-Duo. Could they please explain the rationale of choosing one or the other reporter for the respective experiments?

• Likewise, could the authors please elaborate on why they sampled at different timepoints? While the RNA-seq samples (and most other readouts) were taken at 8 h p.i., the CHQ resistance assay samples were instead harvested at 6 h (Fig. 4) and the samples from the infection assay with Salmonella deletion mutants (Fig. 7) at 7 h p.i.

• Strictly speaking, the use of transcriptional reporters (GFP expression from specific promoters) shouldn’t be considered validation of the RNA-seq, since the former measures promoter activity and the latter steady-state transcript levels (i.e. the sum of de novo transcription and RNA decay). Could the authors please rephrase the corresponding sections? Moreover, may this explain why not all of the selected genes could be “validated” by fluorescence reporter assays (Fig. S6)?

• Fig. 2: The pathway plot is fine. However, I’d appreciate if the authors also included one panel here that visualizes the entire expression dataset for all Salmonella genes (e.g. a volcano plot [fold-change over significance] or an MA plot (fold-change over abundance]). Given that it is a single comparison (vacuolar vs. cytosolic expression), this shouldn’t take up too much space. I’d further appreciate if mRNAs and sRNAs were colored differently in such a plot and genes that were followed up be labeled by name. Finally, in the pathway plot, the deletion mutants denoted at the left hand side should be italicized.

• Fig. 7A: The plot contains data for a sitA/mntH double knockout (which is associated with a slightly reduced fraction of cytosolic bacteria). The two single mutants (delta-sitA and delta-mntH) should be included to this assay to test which of the two genes is responsible for this (subtle) effect.

• I recommend the authors to include a model figure that summarizes the major findings revealed in this work, comparing Salmonella in the SCV and in the cytosol.

• Methods: A section is missing that describes how the cDNA libraries were prepared and how the sequencing was performed (read length, single- or paired-end mode, etc.).

• References: Some of the references seem to be skewed (examples include Ref #116, #122, #128).

Text edits:

• Lines 228-231: The sentence seems grammatically incorrect.

• Line 301: “Compared to untreated cells at Fig 6 h p.i. (…)” Please remove “Fig”.

• Line 346: A verb is missing from this sentence.

• Line 478: “(…) mutants fail to proliferate in the epithelial cytosol” This seems to be an overstatement. As described in my above comment, from the data in Fig. 7 I rather conclude that the mutants are (slightly) less efficient in escaping from the vacuole and entering the cytosol. In contrast, to me these data do not imply that the mutants fail to proliferate.

• Line 501: “S. flexneri does not encode for mgtB.” This statement sounds redundant as already the previous sentence says that.

• Line 578: “(…) deleted for amino acids 4-Fig 66) and (…)” Please correct.

• Line 596: “(…) replacement was by PCR (…)” It seems like a verb is missing.

• Line 749: “(…) confocal microscope was using the (…)”It seems like a verb is missing.

Reviewer #2: 1. The choice of ESBB to deplete cytosolic bacteria (and therefore to enrich the vacuolar population) and WTM to enrich cytosolic bacteria is disputable:

Can the authors comment why they did not sort the cytosolic bacteria using for example their pNF101 plasmid and perform RNA-seq on a sorted population which would have been neater as opposed to result in a mix between cytosolic and vacuolar bacteria (see fig 1D; after 8h pi in presence of WTM, the vacuolar bacteria are still present)? I agree that this fraction seems to be the same in the EBSS treatment and that the comparison between EBSS and WTM treatments provides the authors with their “up cytosol” genes but an RNA-seq on sorted populations would have been better.

2. Line 145 / Line 409. The authors are using two different Gfp (Ova and ASV), one being more stable than the other. Please explain.

3. Line 167 – 169. The authors refer to the WTM treated bacteria as “enriched for vacuolar Salmonella” but contrarily to EBSS where a clear enrichment of the cytosolic fraction is seen, it is not the case for the vacuolar fraction in the WTM condition. There, what is seen is more a depletion of the cytosolic bacteria than an enrichment of the vacuolar bacteria. The authors should therefore rewrite this specific sentence accordingly. In addition, I believe that instead of “enriched for either cytosolic or vacuolar S. Typhimurium by EBSS-or WTM treatment”, we should read “enriched for either cytosolic or vacuolar S. Typhimurium by WTM – or EBSS treatment”.

4. It is unclear to me why the authors chose to compare directly WTM and EBSS conditions rather than compare WTM with untreated; EBSS with untreated and the results of those two comparisons compared to each other. Please explain.

5. I do miss p-values on the panels for each figure, especially in figure 4B but also throughout the manuscript. Differences are sometimes difficult to appreciate and the number of repeats is very low (n=2). Another example of small but significant differences where p-values on the graph are needed, is figure 7 panel A (even though here, the number of repeats (more than 5) makes the data stronger.

6. T0 controls (before infection or even in the ON culture) are missing for the following panels: fig3, lower panel; fig 4, panels A and B; fig 5 lower panel; fig 6 lower panel. If the authors performed those experiments, it would be a plus to display them.

7. Line 275, fig 4B. Did the authors perform any growth curves experiments to access the toxicity of DPI (especially since DPI affects replication, line 298). If DPI itself is toxic for bacteria, it could explain the Gpf accumulation observed in fig 4B. Please comment.

8. Line 290, fig 4A. A non-iron related gene as a control would have been a great addition. Again, if the authors have done such control, it would reinforce the trends observed in panel A after the gradual addition of FAC.

9. Typo mistakes (twice, line 301 and 302 for 8 hpi).

10. Line 346. Is a verb missing?

11. Overall, the hypotheses are all very well supported by the data, with the exception of Mn. I would value the results of a transcriptional fusion only dependent on Mn. I believe the two the authors mentioned have been tested for iron experiments earlier in the manuscript and since those genes are dependent on both iron and Mn, the investigations of the authors stopped there for Mn. Did the authors find an “up cytosolic” gene only dependent on Mn that they could have tested instead?

12. Line 359. Please add a conclusion to this paragraph regarding all the findings on metal in the two niches.

13. Line 428. Please add a reference for the pCHAR-Duo (ASV).

Reviewer #3: Please use this section for editorial suggestions as well as relative minor modifications of existing data that would enhance clarity.

Lines 123-124: “here we define the niche-specific…” the authors only define some of the environments encountered by S. Typhimurium in these niches.

Line 130: the authors mention genetic manipulation of autophagy levels; can they show any data showing that a non-nutritional upregulation of autophagy reduces the fraction of cytosolic bacteria?

Line 139: first appearance of EBSS, please define it here. It is only defined in line 663.

Lines 175-176: how were the Fold-change values defined?

Lines 228-230: the authors highlight the use of GFP as an “ideal reporter to study differential gene expression”, but they don’t explain why they specifically use the gfpmut3variant

Line 380: LpxR is not a hypothetical protein, but a 3’-O-deacylase of the Salmonellalipid A (see Reynolds et al 2006 DOI: 10.1074/jbc.M603527200 and Kawasaki et al 2012 doi: 10.1016/j.bbrc.2012.10.054). Even more, is quite interesting that an enzyme that modifies the lipid A reducing its ability to activate the immune system, is up-regulated in the host cytosol, but the study of this is beyond the scope of this manuscript.

Line 578: looks like there is a typo in the yjjZdeleted amino acids “4-Fig 66”

Figure 3, 5 and 6: to make it more consistent, maybe consider labelling the single channels with the fluorescence shown instead of having a mixture (GFP and STm)

PLOS authors have the option to publish the peer review history of their article (what does this mean?). If published, this will include your full peer review and any attached files.

Reviewer #1: No

Reviewer #2: No

Reviewer #3: No
---

## [Editor Report · Decision Letter 1]

6 Aug 2021

Dear Leigh,

We are pleased to inform you that your manuscript 'Intracellular niche-specific profiling reveals transcriptional adaptations required for the cytosolic lifestyle of Salmonella enterica' has been provisionally accepted for publication in PLOS Pathogens.

Best regards,

Sophie Helaine

Associate Editor

PLOS Pathogens

Raphael Valdivia

Section Editor

PLOS Pathogens

Kasturi Haldar

Editor-in-Chief

PLOS Pathogens

orcid.org/0000-0001-5065-158X

Michael Malim

Editor-in-Chief

PLOS Pathogens

orcid.org/0000-0002-7699-2064
---

## [Editor Report · Acceptance letter]

23 Aug 2021

Dear Dr. Knodler,

We are delighted to inform you that your manuscript, "Intracellular niche-specific profiling reveals transcriptional adaptations required for the cytosolic lifestyle of Salmonella enterica," has been formally accepted for publication in PLOS Pathogens.

Best regards,

Kasturi Haldar

Editor-in-Chief

PLOS Pathogens

orcid.org/0000-0001-5065-158X

Michael Malim

Editor-in-Chief

PLOS Pathogens

orcid.org/0000-0002-7699-2064